# Shoreline Prediction Modelling as a Base Tool for Coastal Management: The Catania Plain Case Study (Italy)

Fx Anjar Tri Laksono [1,2,*] 🆔, Laura Borzì [3,*] 🆔, Salvatore Distefano [3] 🆔, Agata Di Stefano [3] and János Kovács [1] 🆔

1   The Doctoral School of Earth Sciences, Department of Geology and Meteorology, Institute of Geography and Earth Sciences, Faculty of Sciences, University of Pécs, 7624 Pécs, Hungary
2   Department of Geological Engineering, Faculty of Engineering, Jenderal Soedirman University, Purbalingga 53371, Indonesia
3   Dipartimento di Scienze Biologiche, Geologiche e Ambientali, Sezione di Scienze della Terra, University of Catania, 95129 Catania, Italy
*   Correspondence: anjar93@gamma.ttk.pte.hu (F.A.T.L.); laura.borzi@unict.it (L.B.)

**Abstract:** Coastal dynamic is the complex result of multiple natural and human processes, and past and future coastal behavior studies become fundamental to support coastal zone management. However, the reliability of coastal evolution studies is strongly dependent on the analyzed time interval. The longer the period is, the more reliable the past shoreline change analysis and the forecast of shoreline position will be. The present study showed the 50-years shoreline evolution of the Catania coastal plain (Southern Italy), a densely populated area where human-induced processes profoundly changed natural dynamics. Landsat and Sentinel imageries were used to extract shorelines' position over the time between 1972 and 2022 and the DSAS tool was used to calculate the shoreline change rates. The shoreline evolution in 2032 and 2042 was performed by the Kalman filter method, a tool largely applied to forecast short-term shoreline future position. Most of the Catania coastal plain was mainly retreating over the last decades. However, the most significant changes were registered in correspondence with the coastal structures and the river deltas. The reliability of the forecasting model was highly related to the coastal morphology. As such, the lower RMSE values were calculated in correspondence with the uniform coastal subsectors.

**Keywords:** coastal dynamic; shoreline change analysis; shoreline forecast; DSAS; Kalman filter method





## 1. Introduction

Coastal areas are highly dynamic environments, and their behavior is the complex result of multiple processes occurring and interacting on a variety of time and spatial scales. Nowadays, fast changes in coastal dynamics occur most often due to human interventions. Many studies pointed out how coastal man-made works (ports, coastal defense structures) and interventions on the watershed characteristics (e.g., dams, artificial channels) can alter the sediment balance directly affecting the shoreline evolution [1–3]. Despite the significant role of coastal areas in terms of economic, cultural, and social benefits, coastal land use is often planned with insufficient thought for natural hazards, and coastal structures are increasingly exposed to the risk of wave inundation and damage [4,5]. As such, shoreline studies are vital to the early stages of the decision-making process for planned coastal developments to mitigate the potential loss of buildings, infrastructure, and beaches. Several coastal studies aim to quantify the coastal long-term evolution, also at a geological scale, to support the coastal decision-making process in natural hazards management [6–12].

Shoreline position forecasting is a long-debated issue; shoreline evolution modelling that considers hydro-/morphodynamic conditions seems to be the most reliable tool to predict shoreline evolution [13–15]. As such, adopting a longer time scale, high-resolution quality dataset and linear-regression rates application is highly recommended to perform

coastal evolution forecasting [16,17]. Afterwards, equilibrium models become powerful and quick tools to predict future shoreline positions [18–23]. The Kalman filter data assimilation method is a simple and efficient computational tool that couples observed shoreline positions with model-derived positions to forecast a future shoreline position [24,25]. Lately, an extended application of the Kalman filter method is widely used in coastal dynamic studies, whose advantages include the use of data non-uniform in space and time as well as different types of instruments with different noise variances [26–30].

For this paper, the Kalman filter data assimilation method was thus applied through the Digital Shoreline Analysis System [30] to the coastal area of the Catania Plain case study (Italy), a long-wide sandy beach backed by a highly urbanized area subject to increasing and constant development over the last centuries. However, coastal management failed to promptly deal with the issue of coastal erosion and most of the coast is now endangered [3,31]. The shoreline change analysis was performed on a medium-term time range between 1972 and 2022 using Landsat and Sentinel images and the average shoreline evolution trend was estimated by the U.S. Geological Survey software (USA) Digital Shoreline Analysis System (DSAS®, v.5.1) that works in Esri Geographic Information System (ArcGIS) application. The results were compared to previous insights by [31], which investigated the Catania coastal plain beach changes over a long-term period (1169–2011) and forecast the 2021 shoreline position. Hence, the Kalman filter method reliability and application to this kind of coastal area were assessed.

## 2. The Catania Coastal Plain

The investigated area is approximately 20 km long and the coast is N-S oriented, stretching between the Catania Harbour and the Agnone bagni beach and falls within the Catania coastal plain area. The coast can be split into 2 long sectors using the Simeto River mouth as a physiographic feature edge, N-Sector and S-sector. Both sectors can be even more subdivided into 5 subsectors as shown in Figure 1. The Catania plain is a wide Holocene alluvial formation filled by the alluvial-coastal deposits of the Simeto River [32–34]. The Catania Plain is part of the Catania-Gela foredeep, which split the Hyblean Plateau from the over-thrusting Sicilian-Maghrebian Chain [35–42]. The Plain is a depression filled by Upper Pliocene to Quaternary marine sediments with volcanic intercalations [39]. The Simeto River is one of the longest rivers in Sicily and its watershed is the largest by size, 116 km and 4029 km$^2$ respectively [32,43]. The river stream flows into the Ionian Sea and splits the coast into approximately equal segments of 10 km each. The Simeto River represents the main sediment source to the local longshore load, and the San Leonardo River flows a few km south of the Simeto River but provides a low contribution to the sediment supply [43]. However, significant changes in the Simeto River sediment supply have been reported over time, likely related to such human interventions (e.g., dams, weirs) that partly blocked the riverine load [3,32]. The effects of these modifications to the drainage basin affected the Simeto River delta and, in turn, the sediment budget of the Gulf of Catania shoreline. Moreover, due to the oblique wave approach, the Simeto River delta is wave-dominated and shows a strong asymmetrical shape [32]. Such interventions were implemented along the river course and a significant sediment load decrease was registered since the 1950s. As such, sediment supply decrease seems to be a pivotal factor in the coastal dynamics of the Catania Gulf coupled with bad coastal land management [3,32].

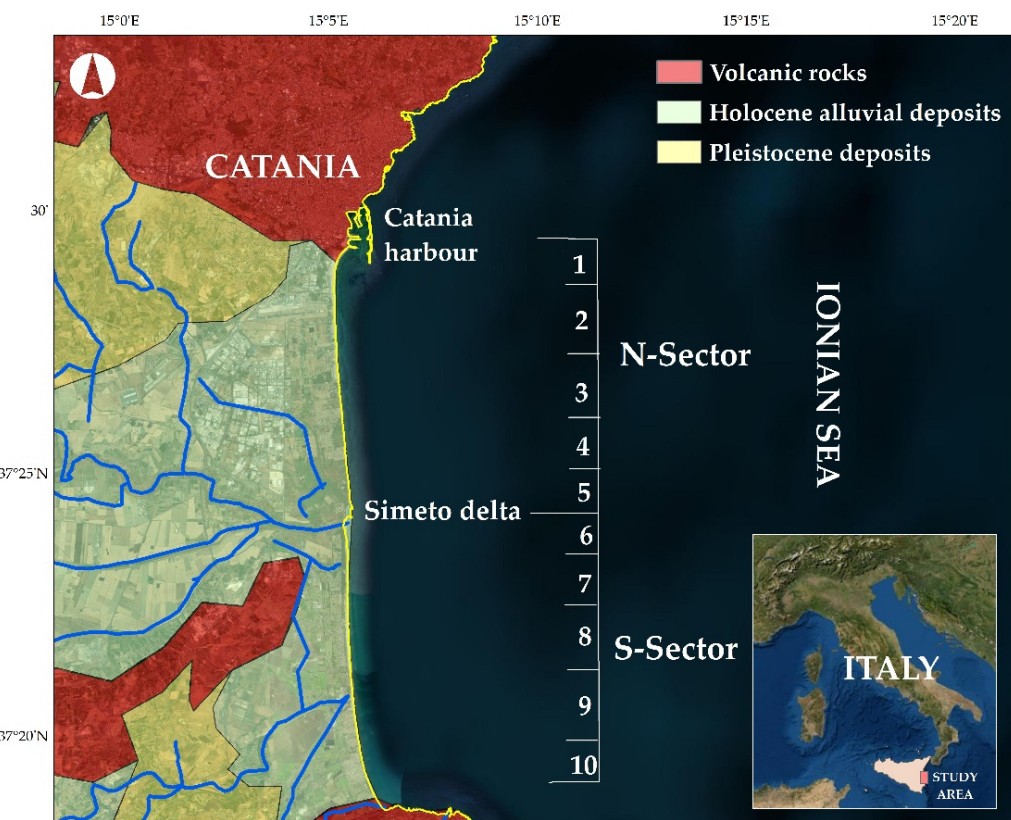

**Figure 1.** The analysed coastal area falls within the alluvial Simeto delta plain (Catania, Southern Italy) that faces the Ionian Sea. The study area was split into two main sectors (N and S) using the Simeto River mouth as the feature edge. The N- and S-sectors were split into subsectors.

## 3. Materials and Methods

### 3.1. DSAS and Shoreline Change Analysis

The shoreline change analysis was performed over a 50-year time interval by using multiple images within the same year at every 10-year interval to ensure the accuracy of the shoreline interpretation. The dataset covered a time span between 1972 and 2022 and included Landsat and Sentinel satellite images (Table 1). Landsat 1–5 Multispectral Scanner (MSS), Landsat 4–5 Thematic Mapper (TM), Landsat 7 Enhanced Thematic Mapper Plus (ETM+), Landsat 8–9 Operational Land Imager (OLI) and Thermal Infrared Sensor (TIRS), and Sentinel-2. Such multiple images in a short period tended to minimize errors in determining the boundaries of water bodies and land (Figure 2) [44–46]. The difference in the time of image capture allows for differences in tidal heights which affect the appearance of shoreline objects in the image. Such errors may occur because differences in the angle and time of image recording will lead to divergence in the position of the coastline [47,48]. Therefore, we identified shoreline objects from different image capture angles to obtain their appearance accurately and comprehensively. We downloaded 210 Landsat and Sentinel-2 images from the United States Geological Survey (USGS) and ESRI. Only imageries with a maximum cloud cover percentage of 20% were used to reduce the shoreline position misinterpretation [44,48]. All downloaded Landsat and Sentinel-2 images were then subjected to radiometric and geometric corrections.

**Table 1.** The Landsat images applied in this study consist of five types with various resolutions ranging from 10 to 80 m. The Green Band and Near Infrared(NIR) are the main bands for determining the boundary between water bodies and land. RT is Revisit Time expressed a number of days [44].

| Satellite | Resolution (m) | Year | Sensors | No. of Bands | Current Status | Green Band | NIR | RT |
|---|---|---|---|---|---|---|---|---|
| Landsat 1–5 | 60 | 1972 1981 1991 | MSS | 4 | Ended in 1992 | Band 4 for Landsat 1–3 MSS. Band 1 for Landsat 4 and 5 MSS. | Band 6 and 7 for Landsat 1–3 MSS. Band 3 and 4 for Landsat 4–5 MSS. | 18 |
| Landsat 4–5 | 30 | 1991 2001 2011 | TM | 7 | Ended in 2013 | Band 2 | Band 4 | 16 |
| Landsat 7 | 15 | 2001 2011 2021 | ETM + | 8 | Operational | Band 2 | Band 4 | 16 |
| Landsat 8–9 | 30 | 2021 | OLI TIRS | 11 | Operational | Band 3 | Band 5 | 16 |
| Sentinel–2 | 10 | 2021 2022 | MSI | 13 | Operational | Band 3 | Band 8 | 10 |

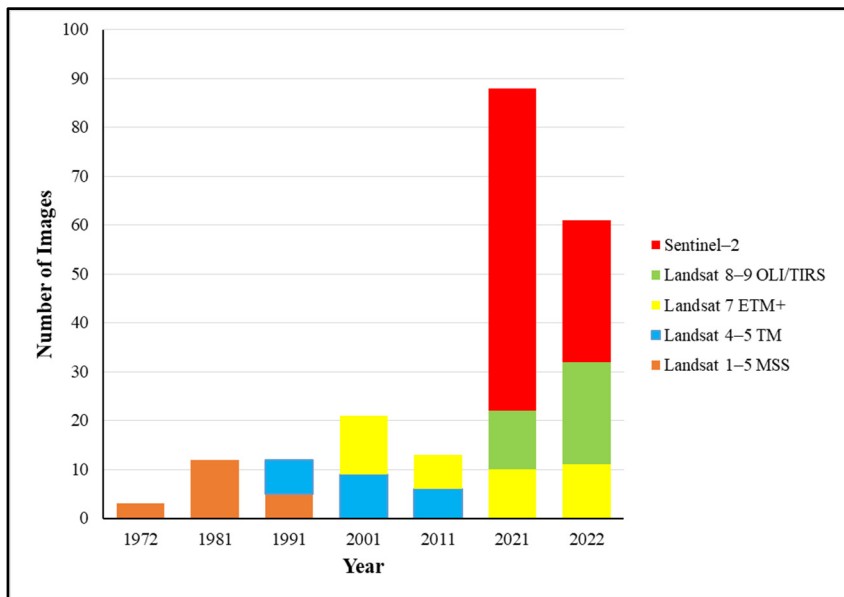

**Figure 2.** Variation in the number of Landsat and Sentinel-2 images adopted in the analysis of Catania coastline change. In 1972 and 1981 there was only one type of Landsat, Landsat 1–5 MSS with three and twelve images, respectively. In 2021 and 2022 there were three types of images analyzed, Landsat 7 ETM+, Landsat 8–9 OLI/TIRS, and Sentinel-2.

The image pre-processing included radiometric and geometric corrections and was performed by the ENVI 5.1 application. Radiometric correction improved the image's visual quality and pixel values by applying both cosmetic and atmospheric corrections. A cosmetic correction was carried out to reduce noise in the images, while atmospheric correction was conducted to enhance the image quality from fog, sun angle, and skylights. In radiometric correction, there is also a dark object subtraction stage that is useful for clarifying the appearance of objects in the image [49,50]. The geometric correction was needed because image data recording is not perpendicular to the object but instead forms a certain angle that causes the actual object's position, shape, and size to be different from the sensor recording results. Geometric correction aims to place remote sensing data in the correct position so that it can be associated with other spatial data. There are two steps in geometric correction, namely spatial interpolation and intensity interpolation. Spatial interpolation is related to the geometric relationship between pixel locations and the earth's surface. This process requires several ground control points (GCPs) that can be obtained from corrected images [49,50]. The next stage was image stacking or combining bands

between short-wave infrared 1 (SWIR 1), near infrared (NIR), and Green [51–53]. SWIR 1, NIR, and Green are light with wavelength ranges of 1.560–1.660 µm, 0.845–0.885 µm, and 0.525–0.600 µm, respectively. Image cropping removes the research location image from the whole image because our case study focuses on the coastline along the south of Catania Harbor to Agnone. The last step was shoreline extraction implementing ArcGIS 10.8.2 until we obtained the final result to separate the water bodies from the land [44,46]. As such, the radiometric and geometric corrected shorelines were compared with shoreline extraction using the normalized difference vegetation index (NDVI), normalized difference water index (NDWI), and the modified normalized difference water index (mNDWI) methods to more accurately delineate the coastline. This method has been implemented by [54–56] to precisely define the land-water boundary in Sindh, Pakistan and the Horn of Africa. NDVI, NDWI, and mNDWI can help to analyze low-resolution imagery so that the possibility of misinterpretation of coastlines is minimized. NDVI was utilized to analyze the shoreline on Landsat 1–5 MSS using band 4 (NIR) and band 2 (red) (Equation (1)). NDVI was applied to the analysis of coastline imagery in 1972 and 1981. The coastline between 1991–2011 was interpreted by the NDWI method adopting band 4 (NIR) and band 2 (green) on Landsat 4–5 TM and Landsat 7 ETM+ (Equation (2)). Meanwhile, the 2021 and 2022 shoreline image investigations tested mNDWI applying band 2 (green) and band 5 (SWIR 1) on Landsat 7 ETM+, band 3 (green) and band 6 (SWIR 1) on Landsat 8–9 OLI/TIRS, and band 3 (green) and band 11 (SWIR 1) on Sentinel-2 (Equation (3)). If the NDVI, NDWI, or mNDWI value is positive, it indicates that the area is the land. On the other hand, their negative values depict water bodies. The boundary between water and land is NDVI, NDWI, and mNDWI, with a value of 0. Shoreline change was also observed by calculating the water frequency index (WFI), the ratio between the number of pixels in the water body, and the sum of the water body pixels and land pixels (Equation (4)) [44].

$$\text{NDVI} = \frac{(\text{NIR} - \text{Red})}{(\text{NIR} + \text{Red})} \tag{1}$$

$$\text{NDWI} = \frac{(\text{Green} - \text{NIR})}{(\text{Green} + \text{NIR})} \tag{2}$$

$$\text{mNDWI} = \frac{(\text{Green} - \text{SWIR1})}{(\text{Green} + \text{SWIR1})} \tag{3}$$

$$\text{WFI} = \frac{N_{\text{water}}}{N_{\text{water}} + N_{\text{land}}} \tag{4}$$

where:

| | |
|---|---|
| NDVI | : the normalized difference vegetation index |
| NIR | : near infrared |
| Red | : visible light with a wavelength of 0.630–0.680 µm |
| NDWI | : the normalized difference water index |
| Green | : visible light with a wavelength of 1.560–1.660 µm |
| SWIR1 | : short-wave infrared 1 |
| mNDWI | : the modified normalized difference water index |
| WFI | : water frequency index |
| Nwater | : the times that the pixels were calculated as water |
| Nland | : the times that the pixels were counted as land |

The reliability of the shoreline extraction from the satellite images was checked by comparing shoreline positions with the shorelines obtained from high-resolution orthophotos (50–100 cm; https://www.sitr.regione.sicilia.it/; http://www.pcn.minambiente.it/mattm/, accessed on 26 April 2022), and significant overlapping between the two datasets was found.

The 20 km coastal sector was split into two mostly equal tracts (ca. 10 km each) as interrupted by the Simeto River delta, the so-called N-sector and the S-sector. The N-sector extends from Acquicella Porto to Primosole Beach, and S-sector stretches from Primosole

Est to Agnone Bagni Beach. The U.S. Geological Survey software (USA) DSAS version 5.1 application was used to assess the main shoreline displacements over the medium-term period by calculating the net shoreline movement (NSM), shoreline change envelope (SCE), end point rate (EPR), and weighted linear regression (WLR) indexes at a CI of 95% (Figure 3). The NSM parameter is the distance between the oldest and youngest shorelines for each transect (Equation (5)) [30]. A negative value of NSM means that the coastline is likely moving landward. On the other hand, a positive value of NSM indicates that the shoreline changes towards the sea. SCE represents the farthest distance between all shorelines intersecting the transect without considering the date of each shoreline and expressed the maximum sediment beach loss (Equation (6)) [30]. The statistical parameter EPR is obtained by dividing the space of shoreline change in a certain period by the time elapsed between the oldest and the latest coastline (Equation (7)) [30]. The EPR parameter was accessible to perform because it only requires at least two shorelines at different times. A positive EPR value reveals that the coastline is accreting. Meanwhile, a negative EPR value implies shoreline recession. The following statistical parameter is WLR (Equation (8)) [30], which means that more reliable data are given greater emphasis or weight to determine the best line. Concerning shoreline change, greater emphasis is attached to data points that have smaller positional uncertainty. The uncertainty field of a shoreline change feature class was employed to calculate the weight.

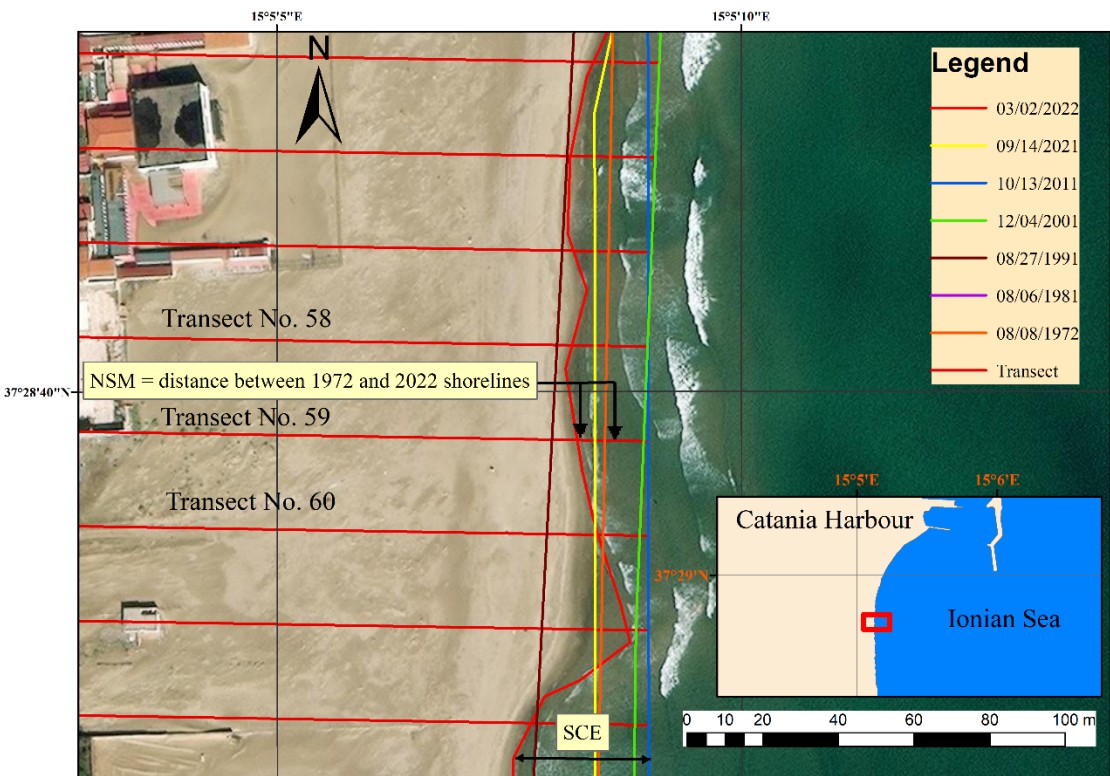

**Figure 3.** Illustration of the transect, statistical parameters of NSM, and SCE in DSAS 5.1. In the figure, NSM represents the distance between the coastlines of 2022 and 1972. SCE represents the furthest distance between two shorelines; in this case, the distance between the 2022 and 2011 shorelines. The image of this illustration was taken from ESRI with a spatial resolution of 1 m, it is much higher than the 30 m and 10 m resolution of Landsat 8 OLI and Sentinel 2. Main figure site location is shown by the red frame in the overview inset.

$$NSM = D_{oldest} - D_{youngest}. \tag{5}$$

$$SCE = D_{max} \tag{6}$$

$$\text{EPR} = \frac{\text{NSM}}{T_{\text{oldest}} - T_{\text{youngest}}} \tag{7}$$

$$\text{WLR} = \frac{1}{e^2} \tag{8}$$

where:

| | |
|---|---|
| NSM | : net shoreline movement |
| D | : Distance |
| SCE | : shoreline change envelope |
| EPR | : end point rate |
| T | : time |
| WLR | : weighted linear regression |

The 1972 and 1981 shoreline determination Landsat images were carried out using the NDVI method by taking the value of 0 as the essential reference for separating water bodies and land areas. NDVI values greater than 0 represent the land area, while NDVI less than 0 indicates water bodies. The utilization of three different approaches (NDVI, NDWI, mNDWI) was due to the different number of bands and resolutions in each type of imagery. The threshold value applied to the NDWI and mNDWI methods is the same as the NDVI method, which is 0. Positive values indicate the land area, while negative values of each parameter reveal water. Zero value selection as a reference for delineating the water and land boundary was made with several trials until accurate results were obtained and represented actual conditions. Subsequently, we compared the NDVI (Figure 4A), NDWI, and mNDWI (Figure 4B) values computed by ArcGIS 10.8.2 with the presence of the Simeto River object in the analyzed image. The Simeto River object was taken as a reference for determining the NDVI, NDWI, and mNDWI values because this object was the most prominent and easily recognizable compared to other objects. Finally, the threshold selection was conducted until we could ensure that the position of Simeto River on the NDVI, NDWI, and mNDWI maps was the same as the Simeto River position on the image.

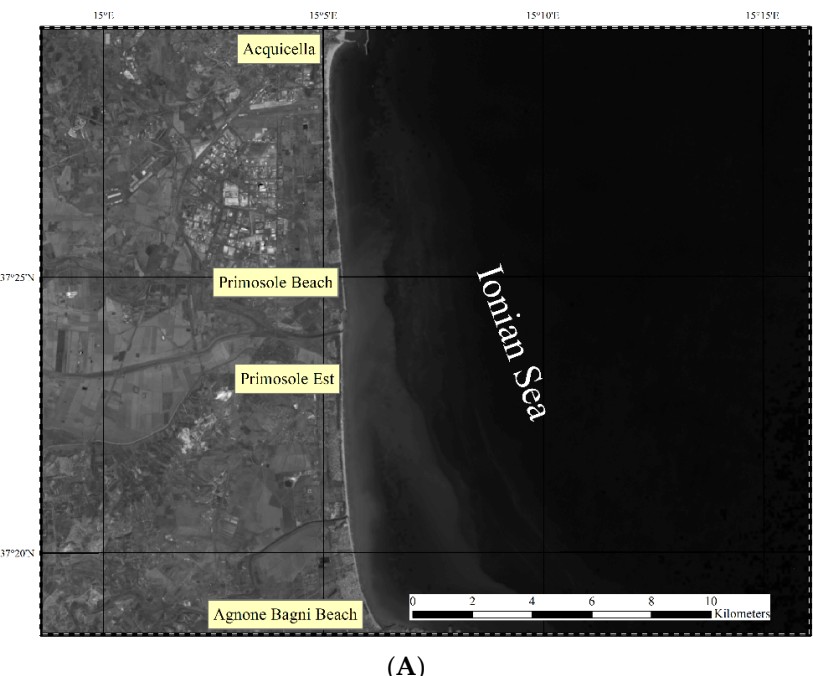

(**A**)

**Figure 4.** *Cont.*

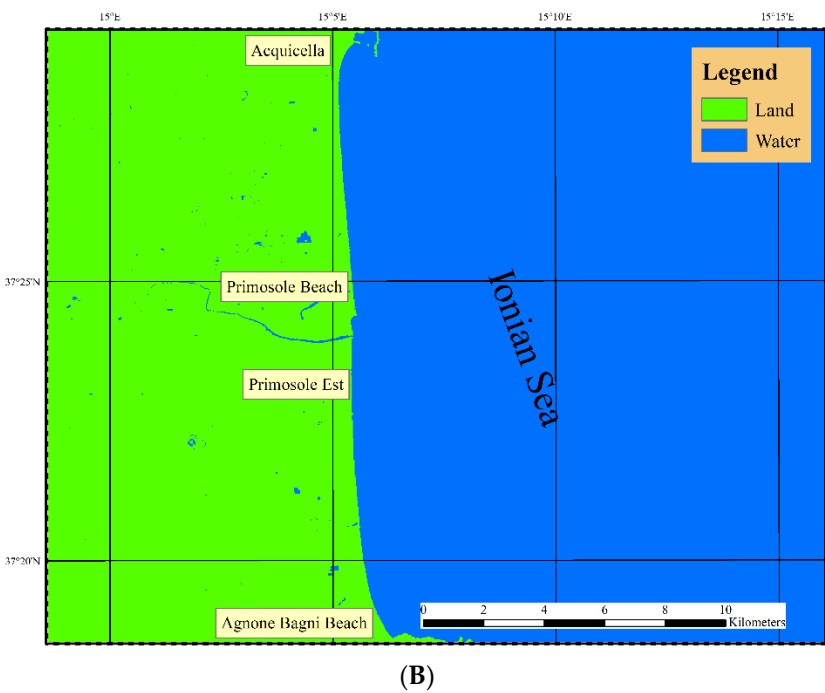

**(B)**

**Figure 4.** Determination of the coastline that divides land and water. (**A**). Catania coastline on 2021 Land-sat-8 OLI. (**B**). Application of mNDWI method on Landsat 8 OLI imagery of 2021 to determine the coastline.

### 3.2. Forecasting Shoreline Position

The results of shoreline change analysis were used to forecast future shoreline position. The Kalman filter method was applied to predict the coastline position in the next 10 and 20 years. The Kalman filter model was recently added as a tool to the DSAS 5.1 and computes the future shoreline position through the observed shorelines considering both the rate and the uncertainty of each one to enhance the reliability of the forecasting model [24–26,28–30]. The linear regression rate (LRR) is needed to run the forecasting computation by DSAS 5.1. The shoreline forecast displayed a polyline shoreline feature class and a point feature class. The point feature class was employed to plot and export the coastline forecast data. The future shoreline forecast in DSAS is based on the input shoreline datum. If all shorelines have the same datum, such as mean-high water (MHW), the resulting shoreline will be the potential location of the future MHW shoreline. If the input data contains multiple shoreline proxies, e.g., MHW and the historical high-water line (HWL), a proxy datum bias must be applied to use shoreline forecasting. The proxy-datum bias (PDB) will convert the HWL shoreline to the MHW datum, and all forecasted shorelines will be referenced to MHW. If the transect does not have a PDB, shoreline forecasting has a built-in solution to apply a regional conversion to the transect so that all forecasted data will be referenced to MHW. Once the calculation rates were carried out and the PDB was solved, the 10- or 20-year shoreline forecast was thus performed by inputs from 1972, 1981, 1991, and 2001 shoreline data to estimate the 2011 and 2021 shorelines. The forecasting process was validated by comparing the forecast 2011 and 2021 shorelines to the extracted 2011 and 2021 shorelines [26,29]. However, a root mean square error (RMSE) was then calculated from the distance between the estimated 2011 and 2021 shoreline horizons and the extracted 2011 and 2021 shoreline position to check the reliability of the Kalman filter model applied to this coastal tract. The maximum RMSE value used to assess the reliability of the Kalman filter model is 60 m, corresponding to the maximum pixel resolution of the satellite datasets. As shown by [29], long and flat coasts provide statistically more reliable results in shoreline forecasting using the Kalman filter method. As such, the Catania coastal plain is a uniform-shape coast, and it could be a good test site to perform shoreline prediction.

## 4. Results

*4.1. Catania Coastal Plain Evolution between 1972–2022*

The medium-term shoreline evolution analysis of the Catania coastal plain was performed by applying the land/water classification of Landsat and Sentinel satellite images to extract shorelines. Evolution rate estimation was carried out using the Digital Shoreline Analysis System (DSAS) working in an ArcGIS environment and covering a 50-year time interval between 1972 and 2022. The coastal stretch was split into two sectors using as feature edge the Simeto River mouth, the N-sector, and the S-southern respectively. The analysis showed that 493 transects on 809 were negative and 316 were positive. As such, 61% of the coast experienced landward migration over the considered time range and only 39% of the shoreline moved seaward. However, shoreline fluctuations of the slight entity were registered within the Catania coastal stretch with 41% of the LRR transects values ranging between ±0.5 m/year. Landward movements between 0.5 and 1.5 m/year occurred in 25.5% of the transects and higher values (more than 1.5 m/year) were reached in 15% of the total no. of transects. Shoreline accretion was recorded for 19% of the transects, even though the most representative changes ranged between 0.5 and 1.5 m/year with 14% of the transects (Figure 5).

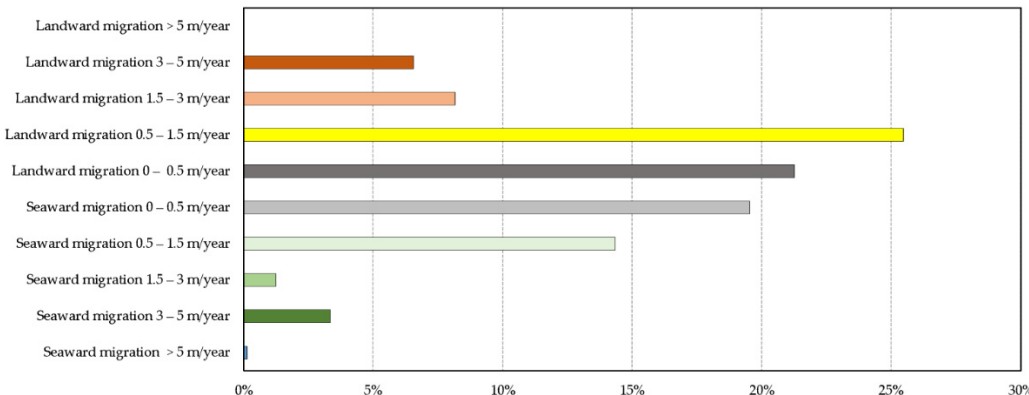

**Figure 5.** Shoreline evolution of the Catania coastal plain between 1972 and 2022. The percentage distribution of the LRR is shown. Most representative ranges are the slight shoreline movements between ±0.5 m/year (41% of the total no. of transects). However, coastal retreats mainly hit the coast with 40% of transects registering negative values lower than 0.5 m/year up to 5 m/year. Slight accretion was also recorded with values ranging between 0.5–1.5 m/year and representing 14% of the coast.

However, the higher accretion rates were observed within the N-sector, in correspondence with the Playa beach (1), south of the Catania Harbour, where the highest LRR was recorded (5.11 m/year). Except for the Playa beach site which faced mainly seaward shoreline movements over the last 50 years, the N-sector was mostly characterized by a slight negative shoreline recession. Afterwards, the recession seemed to gradually increase, moving southward, up to the northern bank of the Simeto River mouth (5), where the highest negative LRR was detected (−4.07 m/year). The coastal area between S. Giuseppe alla Rena and Primosole (4) experienced lower landward migration with rates up to −1.2 m/year. Within the S-sector, two main tendencies were identified: (i) significant shoreline retreats within the coastal stretch between the southern Simeto River mouth (6) and Vaccarizzo beach (7), with higher negative values observed in correspondence with the Simeto River mouth and reaching up to −3.2 m/year; (ii) the coastal area between the S. Leonardo beach (8) and the Agnone Bagni beach (10), where seaward migration was registered with values ranging between 0 up to 1 m/year (Figure 6).

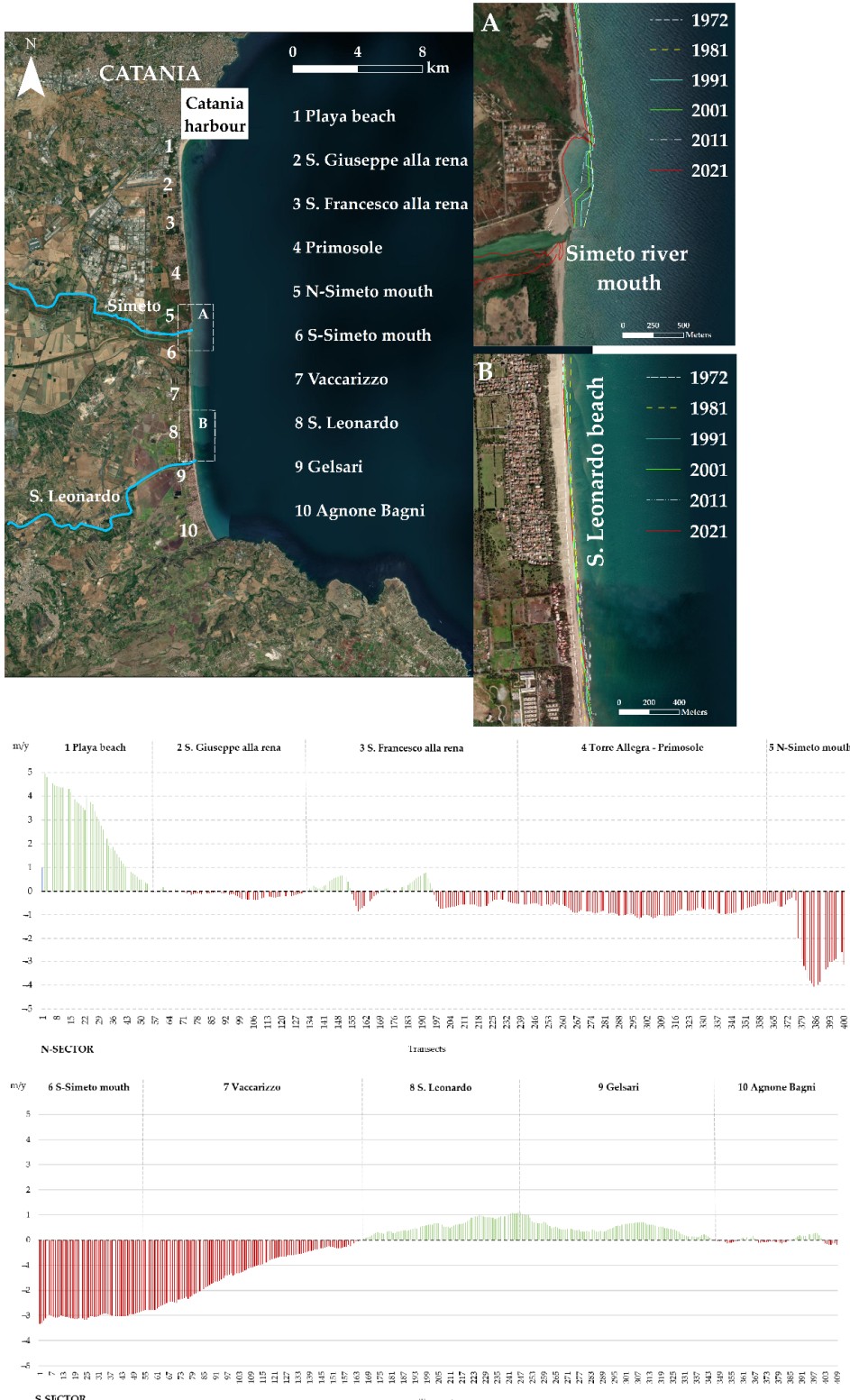

**Figure 6.** The Catania coastal plain area was subdivided into 10 subsectors and the shoreline evolution was expressed by the LRR (m/year). The shoreline evolution analysis showed different migration patterns, (**A**) the Simeto River mouth shoreline faced huge landward migration and a significantly high morphological variability, as also shown by the N-Sector LRR plot; (**B**) the S. Leonardo beach subsector shorelines are partially overlapped, the shoreline migration alternated slight seaward and landward migration, however, the tendency was here slightly positive with LRR values reaching the maximum value of 1.2 m/year, as also shown by the S-sector LRR plot.

The Simeto River mouth (subsectors 5 and 6) profoundly changed over the considered time interval, retreating by up to −200 m as shown by the NSM index (Figure 7). The Playa beach (1) accreted up to 200 m in the northern part of the coastal stretch, south of the Catania harbor.

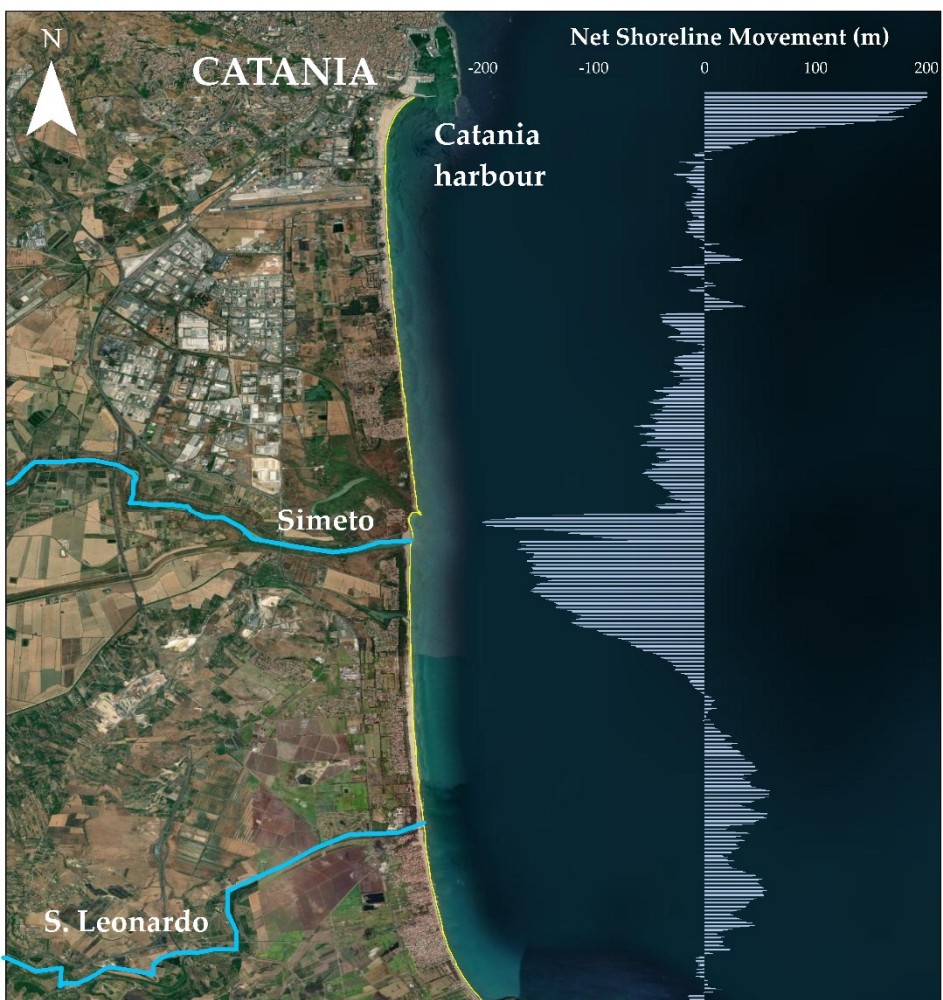

**Figure 7.** The Net Shoreline Movement plot shows the maximum and minimum shoreline displacements. The highest seaward migration reached 200 m in correspondence with the Playa beach, southerly the Catania harbour; the Simeto River mouth experienced the most significant shoreline retreat up to 200 m landward migration.

### 4.2. Catania Shoreline Horizons in 2032 and 2042

The trend of coastline changes between 1972 and 2022 was implemented to predict the Catania shoreline changes in 2032 and 2042. The coastline position forecast in the next 10 and 20 years is based on computing the LRR statistical parameters of the 1972–2022 coastlines simulated by the Kalman Filter method. The Kalman filter method was first tested by forecasting the 2011 and 2021 shoreline horizons by using the 1972–2001 shoreline change rate (LRR). As such, the shoreline obtained from the satellite images of 2011 and 2021 was compared with the 2011 and 2021 predicted shorelines, and the RMSE was computed for each sector and subsector. As a result of this process, the Kalman filter model error was higher in the northern and southern parts of the northern coastal sector (N-sector), but high overlaps were found between the extracted and predicted 2011 shorelines in correspondence with the central parts of the studied coastal sector (from S. Giuseppe alla Rena to Primosole) with RMSE values ranging between 7 and 10 m. However, less reliable RMSE error was found within the N-Simeto River mouth subsector (5). The comparison

between the 2021 extracted and predicted shorelines of the N-sector pointed out less precision than 10-year forecasting, but still within the highest raster pixel resolution in the northern and central parts with RMSE values included between the range 23 and 31 m. As for the 2011 shoreline prediction, the area of the northern bank of the Simeto River mouth showed a high RMSE value. Within S-sector, the Agnone bagni subsector (10) had lower RMSE values, 12.5 and 15 m in 2011 and 2021 extracted and predicted shorelines comparisons, respectively. The highest RMSE values were registered within the S-Simeto River mouth (6) and the S. Leonardo (8) subsectors.

The above-mentioned validation findings were used to properly apply the Kalman filter method to forecast the 2032 and 2042 shorelines. The areas where the RMSE values between 2011 and 2021 extracted and predicted shorelines were higher than 60 m were not included. As such, the Simeto River mouth subsectors (5 and 6) were excluded. Within the N-sector, the Playa beach (1) will register further sedimentation processes moving seaward with a total sediment deposition area of ca. 52,000 $m^2$; alternation of retreating and accreting tendencies was found within the S. Giuseppe alla Rena and S. Francesco alla Rena subsectors (2–3), where a total beach area loss of 14,600 $m^2$ will be registered between the time 2022–2042, whereas Torre Allegra-Primosole coastal tract (subsector 4) will experience a persistent landward shoreline migration with a beach area loss of 68,000 $m^2$ between 2022 and 2042. Within S-sector, Vaccarizzo beach (7) will record a total beach area loss of 94,000 $m^2$, with higher landward movements recorded in the northern part of the subsector and slower recession found in the southern part (Figure 6). A slight accretional tendency will occur within the S. Leonardo subsector (8) with a sediment surface gain of 24,000 $m^2$, the Gelsari subsector (9) will face an alternation of seaward and landward movements, but still, mainly record sediment deposition processes with 10,000 $m^2$ of beach area gain; and the Agnone bagni beach (subsector 10) will register a significant sediment loss of 12,000 $m^2$ wide (Figure 6).

Figure 8A exhibits two different patterns of shoreline changes between Acquicella in transect 65 to 72 and Spiaggia Libera 3 in transect 391–399. In transect 65–72, the shoreline position tends to protrude further seaward in contrast to transect 391–399, which retreats further landward. The 2032 and 2042 shoreline predictions for the Primosole Est-Agnone Bagni Beach segment (Figure 8B) illustrate that Primosole Est between transect 31–42 is eroding and Agnone Bagni Beach between transects 387 and 394 is accreting. The shorelines of 2032 and 2042 form a pattern, coinciding with each other at transect 387–394. Finally, it represents that the coastline is almost stable during this period. The trend of segment-2 is distinct from segment-1, which experienced accretion and erosion throughout the shoreline without the existence of overlapping patterns.

Figure 9 exhibits the forecast uncertainty of 2032 and 2042 shoreline positions in the Acquicella-Primosole Beach and Primosole Est-Agnone Bagni Beach segments. The coastline forecast results in 2032 and 2042 are still within the uncertainty bands, such as the coastline around Primosole Beach (Figure 9A) and Primosole Est (Figure 9B). In both figures, the 2032 coastline is green, while the red color depicts the 2042 coastline. Both coastline positions are still within the light blue- and orange-colored planes in Figure 8A and the dark blue and light green color in Figure 8B. The light blue and dark blue colored planes are the 2042 uncertainty band, while the orange and light green colored planes express the 2032 uncertainty band.

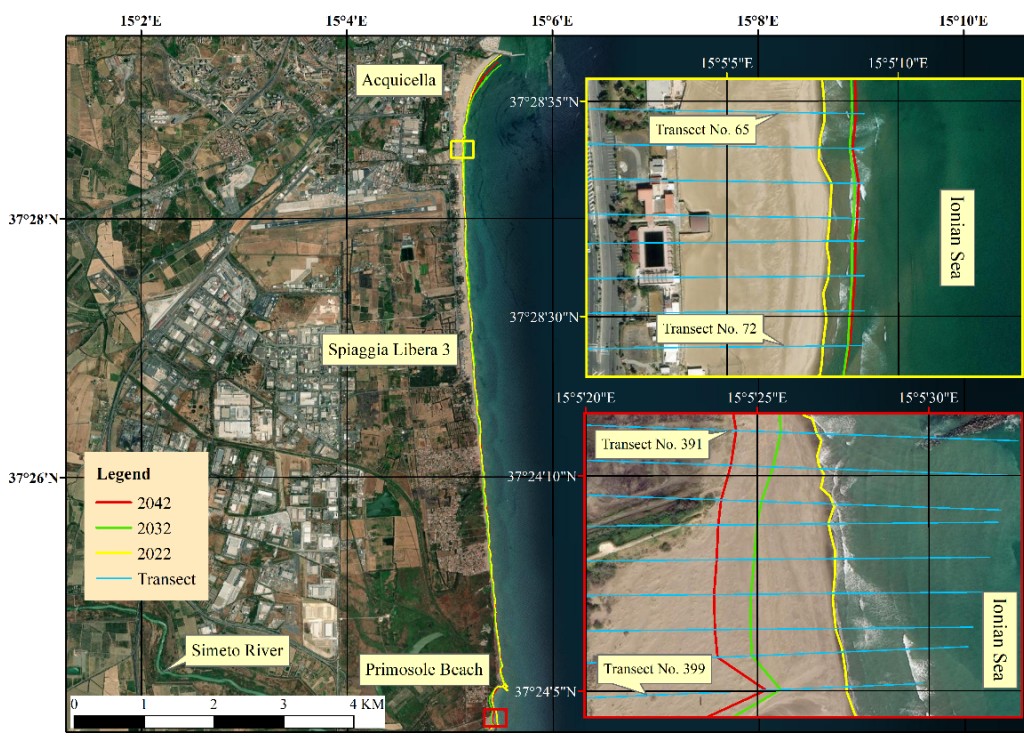

(**A**)

(**B**)

**Figure 8.** Using the Kalman Filter model, the 2032 and 2042 shoreline position changes in (**A**). the Acquicella-Primosole Beach and (**B**). Primosole Est-Agnone Bagni Beach segment can be estimated. At Agnone Bagni Beach, between transect 387–394, there will be accretion in 2022–2032, followed by no significant shoreline changes in 2032 and 2042.

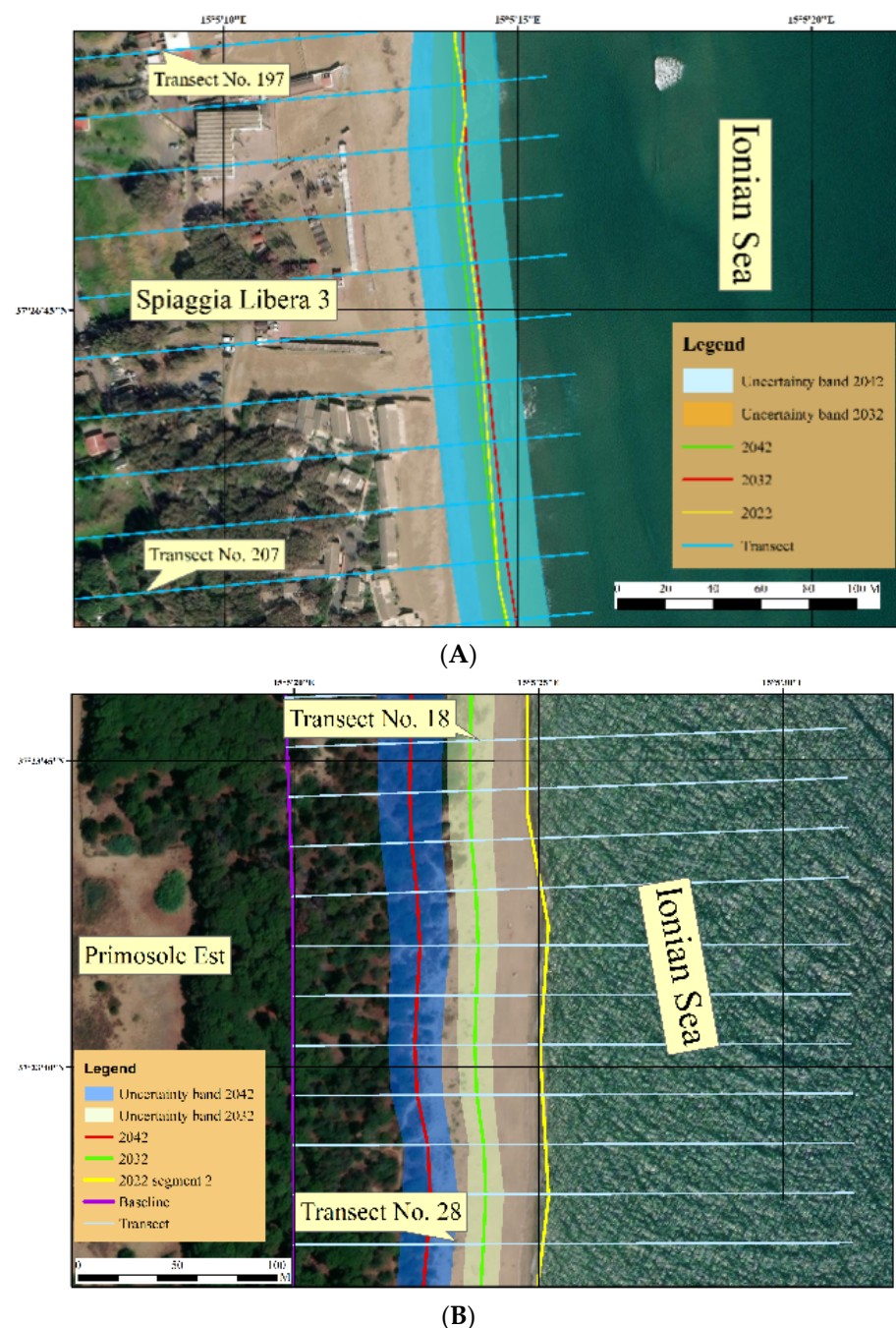

**Figure 9.** Uncertainty of shoreline position change around (**A**). Primosole Beach and (**B**). Primosole Est. Simulated shoreline predictions in 2032 and 2042 are still within the uncertainty band.

Figure 10 illustrates the results of shoreline changes analysis between 2022 and 2042. Based on the results of calculating EPR and WLR statistical parameters, we found similar numbers of transects and percentages for all shoreline change classifications. In 2042, shoreline change in the Acquicella-Primosole Beach segment will be dominated by moderate erosion with a percentage of 47.13% or 189 out of 401 transects. The second largest classification is moderate accretion with a percentage of 31.92% or 128 transects. The percentage and number of transects for each classification of very high accretion, high accretion, very high erosion, high erosion, and stable are 8.73% (35 transects), 6.73% (27 transects), 3.24% (13 transects), 2% (eight transects), and 0.25% (one transect). The number of transects that fall into the erosion category is 210 out of 401 transects, while for the accretion category there are 190 transects. There is only one transect classified as stable in the Acquicella-

Primosole Beach segment based on the calculation of WLR parameters. The estimated average erosion and accretion rates in this segment are −0.598 m/year and 1.334 m/year.

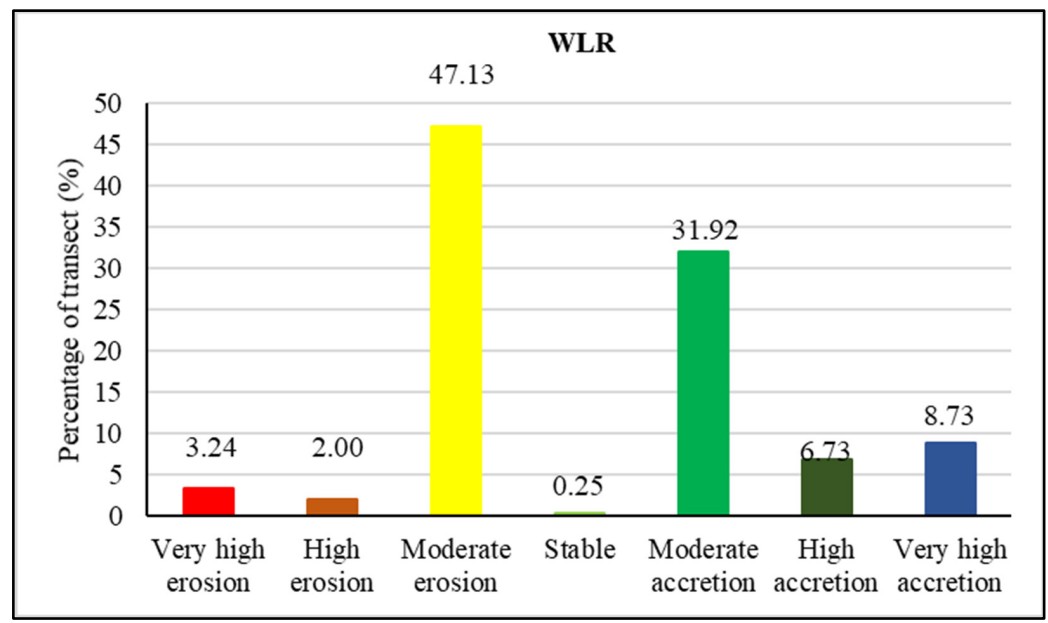

**Figure 10.** The WLR bar chart demonstrates that the predicted rate of shoreline changes at Acquicella-Primosole Beach (N sector) is dominated by moderate erosion with a transect percentage of 47.13%. Out of 401 transects, 189 transects represent this percentage. The eroding shorelines are more dominant than shorelines that are stabilizing or accreting.

The graphical diagram of EPR and WLR as shown in Figure 11A presents a pattern of overlapping. The pattern indicates that the EPR and WLR values at each transect number along the segment are similar. From the diagram, it can also be interpreted that the Acquicella area is dominated by accretion with a maximum rate of 5.75 m/year. Meanwhile, the average accretion rate across the segment was only 1.334 m/year. In contrast to Acquicella, the Primosole Beach area tends to experience the most intensive erosion rates compared to other areas in the segment. The maximum rate for the very high erosion classification reaches −4.48 m/year while the average erosion rate is −0.598 m/year. The most stable area in this segment is throughout Spiaggia Libera 3 with erosion and accretion rates close to 0 m/year. In Figure 11B, which is a graph of NSM and SCE, it can be seen that at Primosole Beach the SCE graph is at the maximum peak while the NSM graph is at the minimum point. The mean, maximum, and minimum values of NSM in this segment are 6.371 m, 114.9 m, and −89.61 m, respectively. At Acquicella, the NSM and SCE values overlap and even on transect 1 the values of both parameters are the same at 114.9 m. At Spiaggia Libera 3, the NSM and SCE graphs tend to be close to 0 which means that they are relatively more stable compared to Acquicella and Primosole Beach.

The WLR bar chart of the Primosole Est-Agnone Bagni Beach segment in Figure 12 expresses that the moderate accretion classification has the highest percentage of 42.787% or represents 175 transects out of a total of 409 transects in this segment. The next highest percentages in order are very high erosion (81 transects), moderate erosion (70 transects), high accretion (41 transects), high erosion (36 transects), and very high accretion (six transects). Meanwhile, no transect in this segment is classified as stable. The total number of transects predicted to experience erosion is 187 transects, while those experiencing accretion reach 222 transects. The average accretion and erosion rates are 0.772 m/year and −1.646 m/year, respectively.

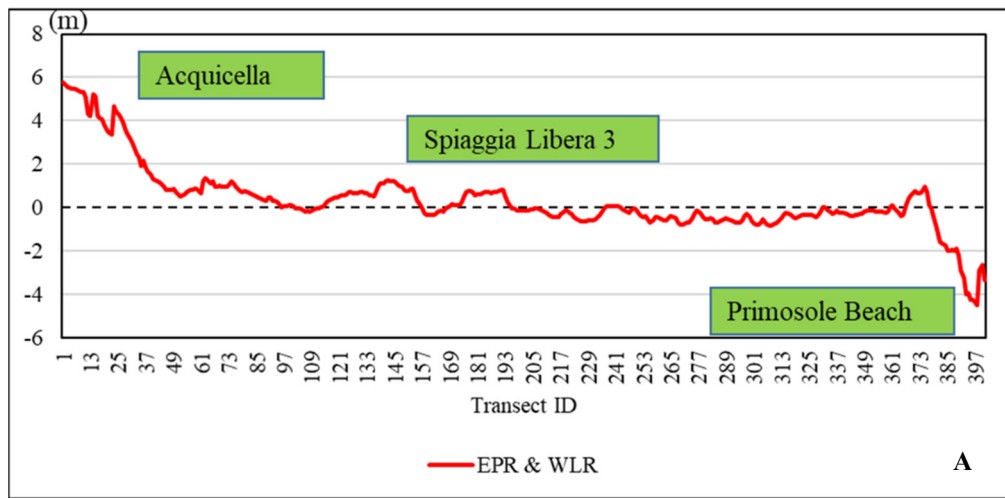

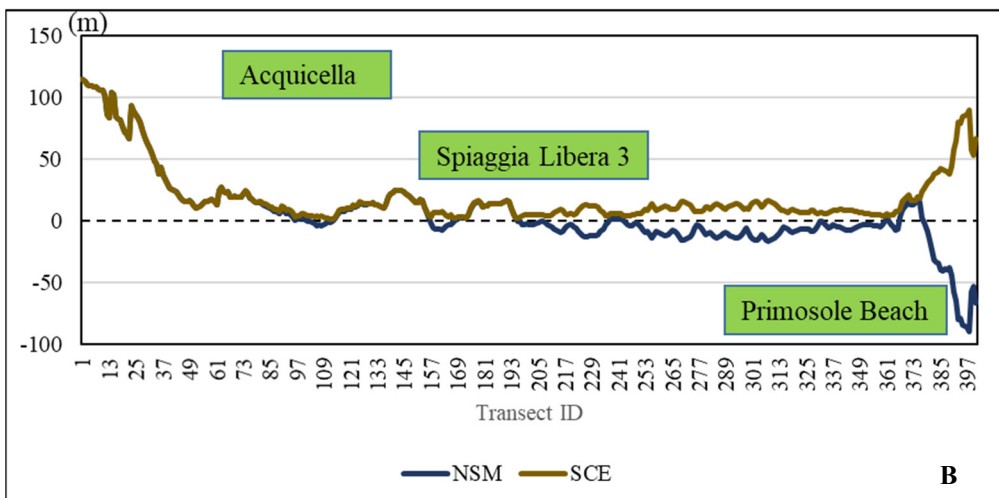

**Figure 11.** (**A**). Shoreline forecasts for the Acquicella-Primosole Beach segment (N sector) in 2032 and 2042 overlap, indicating no differences between the EPR and WLR values. (**B**). According to the NSM and SCE diagrams, Primosole Beach has the most significant erosion, reaching −89.61 m. As compared to the shoreline position in 2022, the Acquicella shoreline experienced the most intense accretion of 114.9 m.

The EPR and WLR graphs of the Primosole Est-Agnone Bagni Beach shoreline prediction in 2042 (Figure 13A) show the same overlapping pattern as the Acquicella-Primosole Beach segment. At Primosole Est, transects 1–155 depict EPR and WLR values below 0 or classified as erosion. Different conditions occur in Via Antille, which tends to be dominated by EPR and WLR values above 0, representing accretion. At Agnone Bagni Beach, the EPR and WLR graphs tend to be close to 0 or can be interpreted as more stable than Primosole Est and Via Antille. Primosole Est is expected to experience maximum erosion at −3.53 m/year, and Via Antille will undergo a maximum accretion rate of 2.14 m/year. The NSM and SCE graphs shown in Figure 13B suggest that the greatest shoreline change will occur at Primosole Est with a minimum NSM of −70.68 m. The maximum NSM value is at Via Antille of 42.87 m while the average NSM is −6.668 m. NSM and SCE graphs overlap at Via Antille, while both graphs are close to zero at Agnone Bagni Beach. This indicator depicts that the shoreline change of the Agnone Bagni Beach segment is relatively more stable compared to Primosole Est and Via Antille.

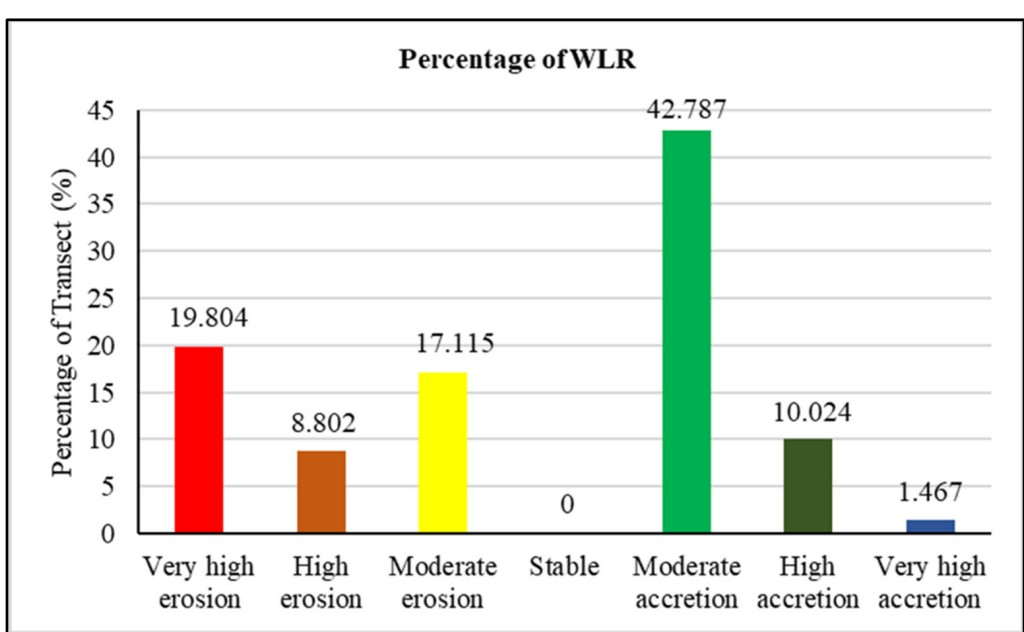

**Figure 12.** Based on the WLRdiagram of the shoreline forecast, the Primosole Est-Agnone Bagni Beach segment (S sector) shows a 42.787% moderate accretion. Overall, there are more transects experiencing accretion than erosion, 222 versus 187. This segment is not expected to have a stable shoreline.

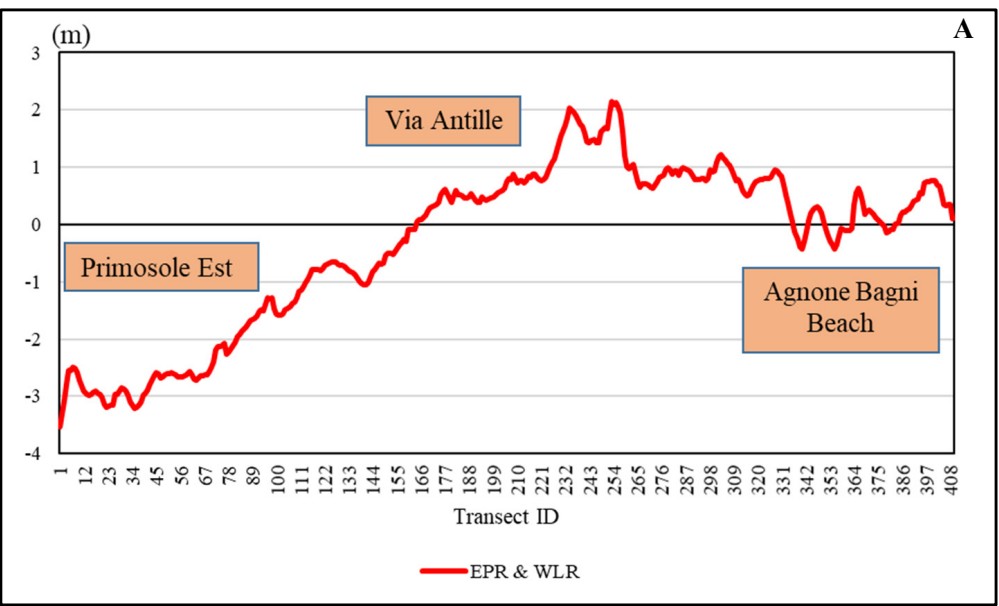

**Figure 13.** *Cont.*

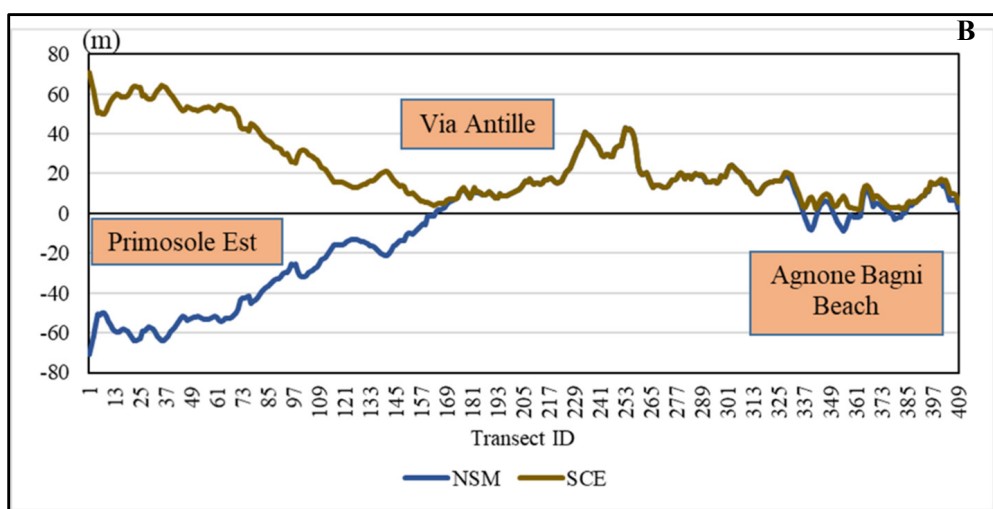

**Figure 13.** (**A**). Based on the EPR and WLR graphs for the Primosole Est-Agnone Bagni Beach segment (S sector), Primosole Est experienced the highest erosion rate of −3.53 m/year. The most significant accretion rate is at Via Antille which reaches 2.14 m/year. (**B**). The NSM and SCE diagrams express that in Primosole Est there is a change in the position of the coastline with the farthest distance between 2022 and 2042 reaching −70.68 m. Meanwhile, Agnone Bagni Beach is relatively more stable than Primosole Est and Via Antille.

## 5. Discussion

*5.1. Simeto Delta and Shoreline Evolution of the Catania Coastal Plain*

The shoreline change analysis performed on the Catania coastal plain over the medium-term of ca. 50 years (1972–2022) pointed out that most of the shoreline faced persistent landward migration with more than 60% of the transects registering negative LRR values [57]. However, 21% of the coast retreated at a slow rate and landward migration between 0 and 0.5 m/year was recorded. Severe landward movements of more than 3 m/year were observed only for 7% of the shoreline, and the highest LRR negative value of −4.07 m/year was detected in correspondence with the Simeto River mouth. Significant accretion rates were estimated within the Playa beach subsector (1) reaching the highest seaward shoreline migration rate of 5 m/year and an average seaward shoreline displacement of 121 m a few meters southward of the Catania harbor (Figure 14). As such, higher sediment deposition rates were observed in correspondence with the up-drift of such coastal structures that trapped sediments of the longshore load, as in the case of the Playa beach subsector, and sediment deposition often induced the up-drift of coastal structures, as already shown by [9,10,58,59]. Erosional processes were recorded in down-drift groins areas, as those emplaced in correspondence with the old Simeto River mouth, which trapped sediments by the longshore drift and blocked the natural lateral migration of the river mouth. Indeed, the highest variability was thus observed at the actual Simeto River mouth as a result of human-induced land modifications (Figure 14). The river mouth profoundly changed over time and the northern bank migrated landward ca. 115 m (average net shoreline movement of subsector no. 5), and the southern bank registered an average landward shoreline displacement of −154 m. The persistent well-known decrease of the riverine sediment supplies due to water capture for human and agricultural use caused a significant loss of beach areas, especially in the subsectors nearby the Simeto delta [3,6].

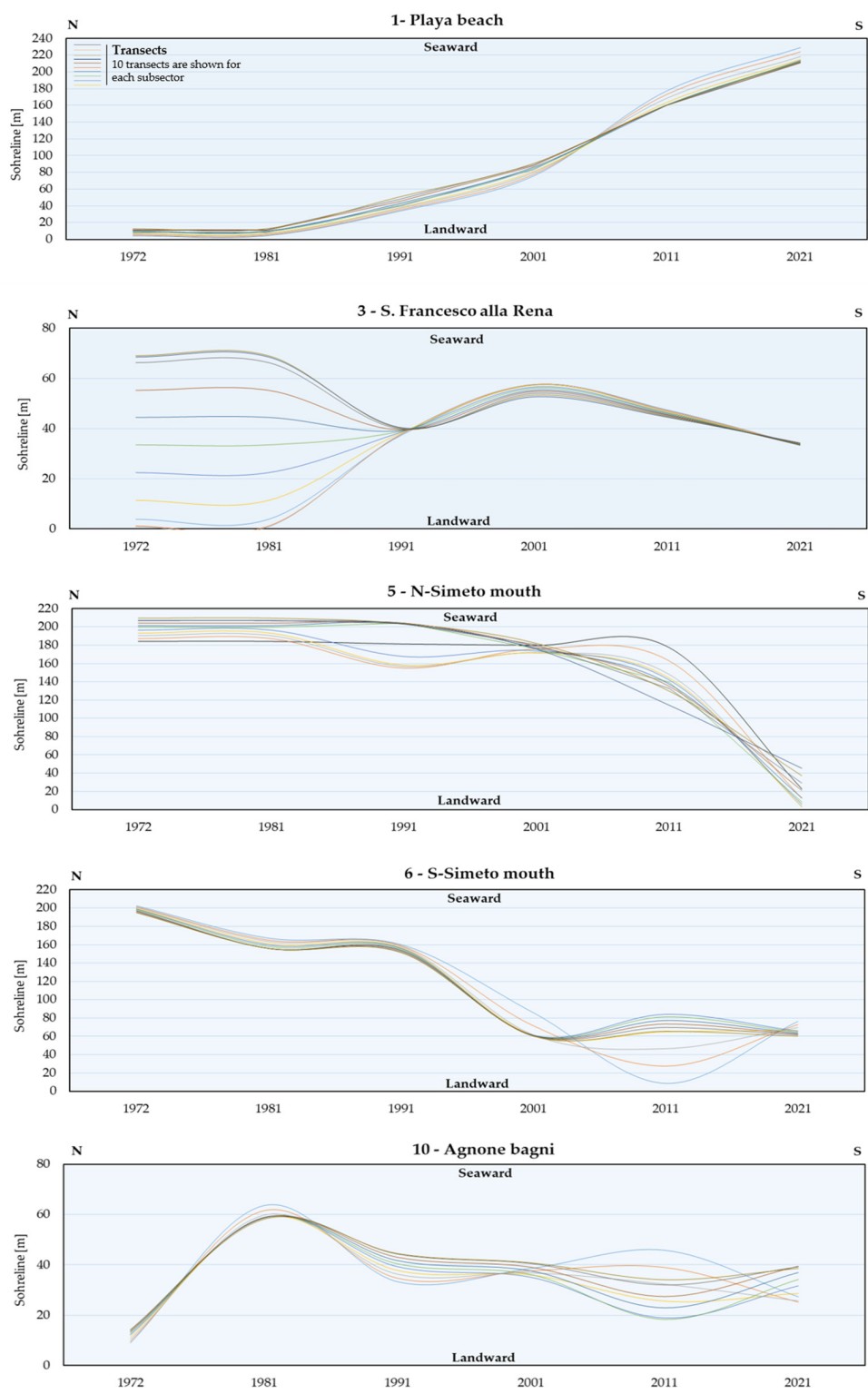

**Figure 14.** Shoreline displacement along the Catania coastal plain sector. The Playa beach subsector is a unique case of significant seaward migration, with n average Net Shoreline Movement of 206 m (average on 10 transects shown). Most of the coast experienced landward migration, even though the most significant shoreline changes were detected in correspondence with the Simeto River mouth. Here, the highest shoreline seaward displacement of more than −150 m was recorded in both river mouth banks (N- and S-Simeto River subsectors, no. 5 and 6 respectively). The central subsectors showed a more stable tendency with an alternation of erosional and accretional phenomena of the slower entity.

Consequently, the river deltas are systems highly sensitive to large- and small-scale environmental changes and such cases are well-documented within the Mediterranean areas [10,60–62]. Other subsectors (2–4, 8–10) showed a uniform and linear morphology N-S oriented and the shoreline evolved with an alternation of landward and seaward migration of lower entity than the Simeto River mouth subsectors (5 and 6) and the Playa beach sector (1).

*5.2. Shoreline Forecasting*

Validation of the 2032 and 2042 shoreline predictions was carried out by comparing the extracted shoreline in 2011 and 2021 with the predicted shoreline in 2011 and 2021. The root mean square error (RMSE) value indicates that the shoreline in the northern segment of the Simeto River in 2011 is between 7.18 and 47.95 while in 2021 it is 23.39–116.39. In the southern segment of the Simeto River in 2011 the RMSE value is between 12.49 and 80.59 while in 2021 it is between 14.72–102.46. Predictions of the Catania coastline in 2032 and 2042 using the EPR, LRR, WLR, NSM, and SCE models reveal the same pattern as the coastline changes between 1972–2022. The same pattern is depicted in the northern sector of the Simeto River and the southern sector of the Simeto River. In the northern sector of the Simeto River, the Acquicella area in 2042 will experience accretion with maximum WLR and NSM reaching 5.75 m/year and 114.9 m, respectively. On the other hand, the Primosole Beach area in 2042 will be subject to erosion with maximum WLR and NSM of −4.48 m/year and −89.61 m, respectively. The overall average rate of erosion across the north of the Simeto River up to 2042 is −0.598 m/year, the average rate of accretion is 1.334 m/year, and the NSM value is 6.371 m. In the southern segment of the Simeto River, the maximum erosion rate up to 2042 occurs at Primosole Est −3.53 m/year while the maximum accretion rate in 2042 is at Via Antille 2.14 m/year. The maximum NSM value is located in Via Antille at 42.87 m, and the minimum NSM occurs in Primosole Est, reaching −70.68 m. Overall along the southern Simeto River the average rate of erosion was −1.646 m/year and the average rate of accretion was 0.772 m. Shoreline changes up to 2042 represented by NSM values state that the coast along the south of the Simeto River will recede by −6.668 m. The large RMSE values, especially in zones near rivers and harbors, are likely to be strongly influenced by factors such as anthropogenic and river sedimentation. In addition, the factors of marine hazard events, sea-level rise, and tidal dynamics are all significant factors in predicting future coastlines so that the accuracy obtained would be perform with higher precision [63–66]. Shoreline prediction using the Kalman filter model method has been applied in several places, such as the coastlines of Oaxaca (Mexico), Göksu Delta and Gulf of Izmit (Turkey), Cuddalore (India), southern Kuwait, the Bay of Bengal (India), and the east coast of India. Moreover, coastline predictions using different methods have also been proposed by [16,27]. Forecasting coastal evolution (ForCE) has been suggested to replace the equilibrium model that are oversimplified and without considering the impact of dynamic sea level, such as tides, waves, and sea level rise [27]. The accuracy of shoreline prediction analysis using LRR is higher than EPR. The LRR parameter was able to reduce the error of shoreline change rate by more than 70% in New York and 34% in Delaware taking into account storm waves, and between 4% and 31% for non-storm data in Delaware and New York, respectively [16]. Shoreline prediction in Oaxaca, Mexico using EPR parameters revealed two distinct patterns of consistent seaward movement in the sector near the mouth of Laguna Chacahua at +0.26 m/year and erosion in the Playa Las Escolleras segment at a rate of −0.45 m/year [26]. Factors affecting shoreline change in Oaxaca, Mexico include rainfall, river discharge, storm, and wave dynamics along estuarine and wetland areas. Meanwhile, anthropogenic factors, such as breakwaters and urbanization, affect shoreline dynamics along harbors and beaches. Maximum fluctuations will occur in the Rio Chacahua Estuary, Verde, Tehuantepec, Playa las Escolleras, and Playa Chipehua coastlines in the next 10–20 years. The shoreline predictions obtained have been validated first by comparing the EPR, LRR, WLR, and NSM values between the extracted shoreline in 2020 and the predicted shoreline in 2020. Predicted shoreline is obtained from

analyzing shoreline changes between 1973–2010 using DSAS 5. followed by calculating RMSE to ensure the accuracy of shoreline predictions [26].

The EPR, LRR, and WLR parameters were compared between the predicted shorelines of the Gulf of Izmit and the Göksu Delta in 2001, 2008, 2009, 2019, and 2020 with the extracted shoreline Landsat images of the same years [29]. Regarding the accuracy assessment of the Göksu Delta shoreline prediction for the next 10 years, the RMSE value of the EPR parameter presented between 0.48 m and 4.2 m. The RMSE value for the LRR parameter is between 0.34–4.04 m, and the RMSE of the WLR parameter is 0.12–1.18 m. The accuracy of shoreline prediction in the Gulf of Izmit in the next 10 years is reflected by the RMSE values for EPR parameters between 1.33–6.78 m, RMSE of LRR parameters between 0.91–6.5 m, and RMSE of WLR parameters ranging from 0.14–1.53 m. The RMSE values of coastline prediction in the next 20 years in the Göksu Delta for EPR, LRR, and WLR parameters are between 0.48–3.44 m, 0.34–3.46 m, and 0.12–2.14 m, respectively. Meanwhile, the RMSE values of EPR, LRR, and WLR parameters in the Gulf of Izmit in the next 20 years are 1.33–4.29 m, 1.04–4.47 m, and 0.14–1.51 m, respectively.

The predicted shoreline position in 2030 at Chinna Vaaikal, India exhibits erosion of 114 m whereas at Puthupettai, India, accretion of 260 m is expected. By 2040, Chinna Vaaikal will experience further erosion of 160 m and Puthupettai will undergo maximum accretion of 278 m. Shoreline erosion at Chinna Vaaikal over the next two decades implies a decrease in the influx of the Coleroon River [28]. A study conducted by Saxena et al. [67] and Natarajan et al. [28] estimated that the Chinna Vaaikal coastline would experience a maximum erosion of −5.5 m/year. Furthermore, the coastlines of Thammanam pettai, Reddiyar pettai, Singarathope, Poochimedu, Kayalpattu, and Devanampattinam will also suffer erosion until 2040. This is also supported by [68] who performed a shoreline change study up to 2012 in Kayalpattu Village which demonstrated high erosion.

Forecasting the southern coastline of Kuwait in 2030 and 2050 was carried out by Aladwani [48] using the EPR parameter of the coastline in 1986–2021. Shoreline changes are likely to reflect erosion and accretion along the southern coastline of Kuwait. The minimum erosion rate is projected to be −0.835 m/year, the average erosion rate is −3.94 m/year, and the maximum erosion rate reaches −8.16 m/year. Meanwhile, the minimum accretion rate is predicted to be 0.665 m/year, the average accretion rate is 4.79 m/year, and the maximum accretion rate is 8.53 m/year. Between 2021 and 2050, it is estimated that the coastline in southern Kuwait will tend stable due to the construction of breakwaters to substitute the existence of beach sand, especially in the Al-Khiran region. The presence of breakwaters will reduce the dynamics of the coastline in the future.

The 2015 and 2025 coastline prediction in the Bay of Bengal, India conducted by Mukhopadhyay et al. [69] depicts that the erosion rate in the northern part of Kusabhadra River estuary and Chandrabhaga Beach is higher compared to the southern part of the coast. On the other hand, the southern part of the coastline near Chilika and Puri is very stable. The shoreline prediction analysis has been validated based on a comparison between the extracted shoreline of 2010 using Landsat imagery and the EPR calculation of the 1972–2001 shoreline change trend. The validation RMSE values were between −3.7–239.41 m with an overall average of 51.34 m. Predicted shorelines on the east coast of India with the LRR model depict an average rate of erosion between 2015 and 2025 of −4.64 m. Meanwhile, between 2015 and 2050 the average rate of erosion increased by −16.25 m. The error value between predicted and actual shorelines is higher in river mouths and deltaic plains. As such, the Catania coastal plain showed how the reliability of the Kalman filter model is quite higher in correspondence with straight coastal subsectors, where the RMSE assessment showed lower value than 60 m. Indeed, the highest RMSE values were detected in correspondence with the Simeto River mouth, which was recognized as a highly dynamic coastal area, and the Playa beach subsector where the coastline showed a slight curvilinear morphology.

## 6. Conclusions

Shoreline change analysis is still a challenging process, but shoreline evolution studies in past and future time are vital tools to support coastal zone management. The Catania coastal plain shoreline evolution has been investigated here on a medium-term time frame between 1972 and 2022. The future prediction of shoreline position is calculated by applying the extended Kalman filter method. The study area was split into two subsectors using the Simeto River delta as feature edge. The DSAS tool was used to perform the statistical analysis computation and to calculate the SCE, NSM, EPR, LRR, and WLR. The shoreline evolution on the medium time interval pointed out that most of the coast faced erosional phenomena, with 61% of the analyzed transects registering negative LRR values. However, 39% of the shoreline experienced seaward migration. Indeed, the higher accretion rates were detected in correspondence with the Playa beach subsector, which is placed down-drift of the Catania coastal harbor, where a 5.11 m/year LRR rate was recorded. Nevertheless, the N-sector was mainly affected by a landward migration tendency that seemed to gradually increase moving southward, up to the northern bank of the Simeto River delta where the highest negative LRR value of −4.07 m/year was registered. The central subsectors showed an alternation of slight recession and accretion tendency with a maximum negative rate of −1.2 m/year. Within the S-sector, significant shoreline landward migration was observed in correspondence with the S-Simeto river bank subsector with −3.2 m/year maximum negative LRR value. The coastal central subsectors registered a slight seaward migration with values ranging between 0 up to 1 m/year. The Catania coastal plain is a densely populated area and water catchment for human and agriculture uses significantly changed the riverine sediment supplies to the coast. As such, the delta system and the most proximal areas showed high shoreline fluctuation with strong retreats. The central parts of studied subsectors are N-S oriented, and coastal morphology is uniform as result of a wave-dominated environment. Afterwards, the shoreline analysis showed an alternation of accretional and erosional phenomena of slower entity than the rates recorded in correspondence with the Simeto River delta. The extended Kalman filter method was thus used to predict 2032 and 2042 shoreline positions. The forecasting model was firstly tested by applying the extended Kalman filter model to predict the 2011 and 2021 shorelines. The forecasted shorelines were compared to the extracted 2011 and 2021 shorelines and the RMSE was computed. The lower RMSE values were thus found in correspondence with the subsectors that had a linear coastal geomorphology, with values ranging between 7 and 30 m for the 2011 forecast shoreline and 12–60 m for the 2021 forecast shoreline. The 2032 and 2042 shoreline forecast pointed out that the central subsectors of the N-sector will experience a maximum net shoreline movement up to 115 m. The highest negative net shoreline movement was found within the S-sector with values reaching 70 m landward migration. As shown by other authors, the reliability of the Kalman filter method increased when applied to normal coastal morphologies, and as such lower RMSE values were obtained in the central subsectors within both the studied coastal sectors.

**Author Contributions:** F.A.T.L. proposed the idea, collected data, performed simulations, interpreted the data, and wrote the manuscript; L.B. performed simulations, analyzed the data, and wrote the manuscript; S.D. modified the idea, interpreted the data, and wrote the manuscript; A.D.S. provided critical notes and managed the research; J.K. reviewed and revised the manuscript and managed the research. All authors have read and agreed to the published version of the manuscript.

**Funding:** The research was funded by UNKP program with research grant number UNKP-22-3-1 for 2022/2023.

**Institutional Review Board Statement:** Not applicable.

**Informed Consent Statement:** Not applicable.

**Data Availability Statement:** Not applicable.

**Acknowledgments:** The authors would like to thank UNKP for providing a research grant to conduct this research. This research was also supported by the University of Pécs, Hungary and the University of Catania, Italy in a research collaboration program. We would like to thank the reviewers who have provided suggestions for improving the manuscript of this article.

**Conflicts of Interest:** The authors declare no conflict of interest.

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
