# Peer review of "Shoreline Prediction Modelling as a Base Tool for Coastal Management: The Catania Plain Case Study (Italy)"

_jmse, doi:10.3390/jmse10121988_

Round 1
Reviewer 1 Report
This manuscript studied the shoreline evolution of the Catania coastal plain (Southern Italy) over past 50 years based on Landsat and Sentinel imageries, calculated the shoreline change rates through the DSAS tool, and further forecasted future shoreline positions in 2032 and 2042 by the Kalman filter method. The reliability of the forecasting model was also discussed. The Catania Plain case study shows the applicability of this approach to relatively straight shorelines. On the whole, the structure of the paper is complete and the work is meticulous.
There are several minor issues in the paper that need to be checked:
1. Line 130, (Add references) please add references.
2. Please check Figure 4. This figure lacks the Figure 4C and Figure 4D described in the text.
3. Line 253-254, the percentages of 69% and 31% should be approximately 61% (493/809) and 39% (316/809).
4. I suggest that the LRR plots in Figure 6 use the same scale to better compare the N-Sector and S-Sector.
5. In Conclusions, the forecasted shoreline positions should be in 2032 and 2042, not 2031 and 2041.
Author Response
Reviewer: Line 130, (Add references) please add references
Response: We agree with the reviewer's opinion regarding the addition of references on line 130. In the revised manuscript line 130 becomes 149. We included references [44,46] on behalf of Quang, D.N.; Ngan, V.H.; Tam, H.S.; Viet, N.T.; Tinh, N.X.; Tanaka, H. Long-Term Shoreline Evolution Using 726 DSAS Technique: A Case Study of Quang Nam Province, Vietnam. J. Mar. Sci. Eng. 2021, 9, 1124 and Nithu R.; Gurugnanam, B.; Sudhakar, V.; Francis, P.G. Estuarine Shoreline Change Analysis Along the Ennore 730 River Mouth, South East Coast of India, Using Digital Shoreline Analysis System. Geod. Geodyn. 2019, 10, 205-212.
Reviewer: Please check Figure 4. This figure lacks the Figure 4C and Figure 4D described in the text.
Response: we have revised this section by removing Figure 4C and Figure 4D from the text. Previously, there was an error in writing so that Figure 4C and Figure 4D appeared in the text but there were no figures. We would like to confirm that there are only Figures 4A and 4B in this manuscript.
Reviewer: Line 253-254, the percentages of 69% and 31% should be approximately 61% (493/809) and 39% (316/809).
Response: We agree with the reviewer on this point. We have revised the beach change classification percentages from 69% and 31% in the previous version to 61% and 39% in the latest version. In the previous version, this was in lines 253-254 while in the latest version of the manuscript it is in lines 283-284.
Reviewer: I suggest that the LRR plots in Figure 6 use the same scale to better compare the N-Sector and S-Sector.
Response: We have revised Figure 6 so that the scales in the N-sector and S-sector are the same.
Reviewer: In Conclusions, the forecasted shoreline positions should be in 2032 and 2042, not 2031 and 2041.
Response: We agree with the reviewer's opinion regarding the misspelling of the shoreline prediction year. In the conclusion section, we have revised the predicted coastline positions in 2031 and 2041 to 2032 and 2042.
(Please see the attachment)

Reviewer 2 Report
comments on "Shoreline Prediction Modelling as a Base Tool ... " by Laksono et al.
The work is about using satellite imagery from Catania Plain to obtain shorelines from 1972 to 2021 and make predictions, using Kalman filter, for the shoreline position in 2032 and 2042.
General comments
In general, the paper is well written and not hard to follow. In spite of that, i think that some sentences require rephrasing (perhaps due to the English, which is not my mother language anyway). Also, i find that a brief description of the capabilities and limitations of DSAS is due. The fundamentals of the application of Kalman filter as it is used should be understood by the reader: I dont see it clear how it is applied in prediction, being familiar with Kalman filter. The limitations of its use should also be clear to the reader. I miss also an estimate of the errors (in meters) that can be for the shoreline: a validation with other sources (in 2011 and 2021 at least)
Other comments
.- please check the whole document for typos or other errors (I found an "Add references" in the manuscript...)
.- define SWIR when introduced
.- around line 134: explain of the if the quality of the different sources is comparable
.- around line 201: this informations has already been given
.- Figure 3: mention the source of the image in the caption (and stress that the resolution is, by far, much better than Sentinel-2 and Landsat)
.- you consider 4 shorelines (1972, 1981, 1991, 2001) and 2011 and 2021 afterwards. How do you actually obtain these shorelines from the bunch og images availble (Figure 2)? Do you use jus one image for each time?. What are the estimated errors?
.- around line 308: RMSE is obtained for each sub-sector?. I can now figure out what is the RMSE considered, but it was not evident when introduced.
.- around line 335: i miss a figure illustrating what you are introducing in the text. Or referring to Figures 11 and 12 already here.
.- Figures 11 and 12: make it clear in the caption that it is N or S sector. The ID is local, not global. I miss the units in the vertical axis.
.- around line 606 (but, in fact, before): how do you define the transects? are they always "shore-normal" or are the parallel?. If parallel, does it have an impact on the results (i guess yes)?
Author Response
Reviewer: please check the whole document for typos or other errors (I found an "Add references" in the manuscript...)
Response: We agree with the reviewer's opinion regarding the addition of references on line 130. In the revised manuscript line 130 becomes 149. We included references [44,46] on behalf of Quang, D.N.; Ngan, V.H.; Tam, H.S.; Viet, N.T.; Tinh, N.X.; Tanaka, H. Long-Term Shoreline Evolution Using 726 DSAS Technique: A Case Study of Quang Nam Province, Vietnam. J. Mar. Sci. Eng. 2021, 9, 1124 and Nithu R.; Gurugnanam, B.; Sudhakar, V.; Francis, P.G. Estuarine Shoreline Change Analysis Along the Ennore 730 River Mouth, South East Coast of India, Using Digital Shoreline Analysis System. Geod. Geodyn. 2019, 10, 205-212. In addition, we have corrected typos throughout whole the text.
Reviewer: define SWIR when introduced
Response: We agree with the reviewer on the importance of explaining SWIR in the text. We have added an explanation of SWIR in lines 142-143.
Reviewer: around line 134: explain of the if the quality of the different sources is comparable
Response: We agree with the reviewer's statement so we have added an explanation on lines 151-154 of the revised manuscript. This method has been implemented by [54–56] to precisely define the land-water boundary in Sindh, Pakistan, and the Horn of Africa. NDVI, NDWI, and mNDWI can help to analyze low-resolution imagery so that the possibility of misinterpretation of coast-lines is minimized.
Reviewer: around line 201: this information has already been given
Response: We agree with the reviewer to remove the double information in the text. In the latest version of the manuscript, we removed the information in lines 224-228.
Reviewer: Figure 3: mention the source of the image in the caption (and stress that the resolution is, by far, much better than Sentinel-2 and Landsat)
Response: The image of this illustration was taken from ESRI with a spatial resolution of 1 m, it is much higher than the 30 m and 10 m resolution of Landsat 8 OLI and Sentinel 2.
Reviewer: you consider 4 shorelines (1972, 1981, 1991, 2001) and 2011 and 2021 afterwards. How do you actually obtain these shorelines from the bunch of images available (Figure 2)? Do you use just one image for each time? What are the estimated errors?
Response: We used multiple images each time. For example, in 1972 we analyzed the coastline using 3 Landsat 1-5 MSS images. In 2021 we analyzed 10 Landsat 7 ETM+ images, 12 Landsat 8-9 OLI/TIRS, and 66 Sentinel-2 images. We evaluated each image based on less than 20% cloud cover and performed a series of corrections such as radiometric and geometric corrections. In addition, we also conducted NDVI, NDWI, and mNDWI analysis to ensure the accuracy of the coastline before the investigation of coastline change with DSAS.
Reviewer: around line 308: RMSE is obtained for each sub-sector? I can now figure out what is the RMSE considered, but it was not evident when introduced.
Response: The RMSE has been explained earlier in lines 266-274. However, a Root Mean Square Error (RMSE) was then calculated from the distance between the estimated 2011 and 2021 shoreline horizons and the extracted 2011 and 2021 shoreline position to check the reliability of the Kalman filter model applied to this coastal tract. The maximum RMSE value used to assess the reliability of the Kalman filter model is 60 m, corresponding to the maximum pixel resolution of the satellite datasets. As shown by [29], long and flat coasts provide statistically more reliable results in shoreline forecasting using the Kalman filter method. As such, the Catania coastal plain is a uniform-shape coast and it could be a good test site to perform shoreline prediction.
Reviewer: around line 335: i miss a figure illustrating what you are introducing in the text. Or referring to Figures 11 and 12 already here.
Response: This text explains the shoreline dynamics referred to in Figure 6.
Reviewer: Figures 11 and 12: make it clear in the caption that it is N or S sector. The ID is local, not global. I miss the units in the vertical axis.
Response: We agree with the reviewer's opinion to add the N or S sector position in the caption of Figures 11 and 12 and to add the unit of meters on the vertical axis.
Reviewer: around line 606 (but, in fact, before): how do you define the transects? are they always "shore-normal" or are the parallel? If parallel, does it have an impact on the results (i guess yes)?
Response: The transect represents a perpendicular line between the shoreline and the baseline. In this study, to simplify and maximize transect creation, we used DSAS in ArcGIS. The distance of each transect is 25 m. Every transect was utilized as a reference in the calculation of EPR, LRR, WLR, NSM, and SCE.
"Please see the attachment"

Author Response
Reviewer: The colors of the legend do not fit the colours in the map. Ex: there is no yellow in the map.
Response: The color of the legend and the image in Figure 1 are the same, it might look different due to the map under the layer.
Reviewer: twelve not 12
Response: We have revised the number 12 in the caption of Figure 2 to the word ‘twelve’.
Reviewer: please check the whole document for typos or other errors (I found an "Add references" in the manuscript...)
Response: We agree with the reviewer's opinion regarding the addition of references on line 130. In the revised manuscript line 130 becomes 149. We included references [44,46] on behalf of Quang, D.N.; Ngan, V.H.; Tam, H.S.; Viet, N.T.; Tinh, N.X.; Tanaka, H. Long-Term Shoreline Evolution Using 726 DSAS Technique: A Case Study of Quang Nam Province, Vietnam. J. Mar. Sci. Eng. 2021, 9, 1124 and Nithu R.; Gurugnanam, B.; Sudhakar, V.; Francis, P.G. Estuarine Shoreline Change Analysis Along the Ennore 730 River Mouth, South East Coast of India, Using Digital Shoreline Analysis System. Geod. Geodyn. 2019, 10, 205-212.
Reviewer: Exclude "Meanwhile"
Response: We have removed the word "Meanwhile" from the caption of Figure 3.
Reviewer: "coastline" not coastlines
Response: We have removed the word "between" in Figure 5.
Reviewer: legend colours do not fit, despite overlaping
Response: We agree with the reviewer's opinion that the color of the legend does not match Figures 11A and 13A. Therefore, we have revised it.
Reviewer: recorded
Response: We have modified ‘recor-ded’ in the caption of Figure 14 to the recorded word.
Reviewer: Avoid starting a phrase with a reference
Response: We have changed the sentence structure to avoid citing references at the beginning of the sentence.
Reviewer: references?
Response: We have added references on lines 564 and 571 of the latest manuscript with reference numbers [16] and [26].
"Please see the attachment"

Reviewer 4 Report
I have carefully read and analyzed the paper. It deals with shoreline modeling on a case study beach in Catania plain. Furthermore, the paper shows the ability of the Kalman filter method to forecast short-term shoreline position, while the shoreline dataset was extracted from Landsat and Sentinel images. However, there is not a significant amount of originality as the already established methods for shoreline analysis are simply used on a new beach.
The grammar and spelling could be improved.
Introduction:
L55: Move the begging of the sentence „For this paper, …“ into the next line.
L100-L105: Are the authors stating that multiple images were taken inside the same year at and ten year interval? Please make this paragraph clearer regarding the image capturing method.
L123-L125: The author states that geometric correction was applied, but does not provide methodology or references for the methodology applied. Please add this.
L129: I assume you were using DSAS 5.0 inside ArcGIS, not only ArcGIS itself ?
L149: What is ‘Green’ in equations 2 and 3. Please add this.
L130: There is ‘(add refereces)’. Please correct this.
L206: There is no Figure 4D. Please correct this.
L293: Why was there strong accretion at the Playa beach and retreat at the Simeto River mouth? Do you have some explanation?
Section 4.1. The authors are commenting on beach changes that are mostly in the range 20-30 m for the period 1972-2021 (about 0.5 m/year) but the resolution of the satellite images is less than the aforementioned changes, for example Landsat 1-5 -> 60-80 m or Landsat 8-9 -> 30-100 m. How reliable are these observations?
Section 5.2. The stated RMSE are rather large when compared to the changes that are predicted. If the common error (RMSE) is of the order of magnitude (up to 116 m in the NS) of the forecasted changes (for example, 114.9 m in the Acquicella area). What does this say about the reliability of the forecasts? Please explicitly compare and comment the accuracy of other studies with the accuracy of this study: “Shoreline prediction using the Kalman Filter model method has been applied in several places such as the coastlines of Oaxaca (Mexico), Göksu Delta and Gulf of Izmit (Turkey), Cuddalore (India), southern Kuwait, the Bay of Bengal (India), and the east coast of India.” Consider normalizing the RMSE with the average beach change to get NRMSE for each beach for easier comparison.
Author Response
Reviewer: L55: Move the begging of the sentence „For this paper, “ into the next line.
Response: We agree with the reviewer's statement and we have made revisions to move the words 'For this paper' to the next line (L55-L56).
Reviewer: L100-L105: Are the authors stating that multiple images were taken inside the same year at and ten year interval? Please make this paragraph clearer regarding the image capturing method.
Response: We took multiple images within the same year at every 10-year interval to ensure the accuracy of the shoreline interpretation. We have also added a description of the image capturing method on lines 98-113 ‘The difference in the time of image capture allows for differences in tidal heights that affect the appearance of shoreline objects in the image. Such errors may occur because differences in the angle and time of image recording will lead to divergence in the position of the coastline [47, 48]. Therefore, we identified shoreline objects from different image capture angles to obtain their appearance accurately and comprehensively. We downloaded 210 Landsat and Sentinel-2 images from the United States Geological Survey (USGS) and ESRI. Only imageries with a maximum cloud cover percentage of 20% were used to reduce the shoreline position misinterpretation [44,48]. All downloaded Landsat and Sentinel-2 images were then subjected to radiometric and geometric corrections.
Reviewer: L123-L125: The author states that geometric correction was applied, but does not provide methodology or references for the methodology applied. Please add this.
Response: We have updated the explanation and reference to geometric corrections on lines 136-141 of the latest version of the manuscript. ‘Geometric correction aims to place remote sensing data in the correct position so that it can be associated with other spatial data. There are two steps in geometric correction, namely spatial interpolation and intensity interpolation. Spatial interpolation is related to the geometric relationship between pixel locations and the earth's surface. This process requires several Ground Control Points (GCPs) that can be obtained from corrected images [49,50].’
Reviewer: L129: I assume you were using DSAS 5.0 inside ArcGIS, not only ArcGIS itself ?
Response: We employed the DSAS 5.0 application which is an additional application that can be installed on ArcGIS and is required for shoreline change analysis.
Reviewer: L149: What is ‘Green’ in equations 2 and 3. Please add this.
Response: ‘SWIR 1, NIR, and Green are light with wavelength ranges of 1.560-1.660 µm, 0.845-0.885 µm, and 0.525-0.600 µm’ (L142-L144).
Reviewer: L130: There is ‘(add refereces)’. Please correct this.
Response: We agree with the reviewer's opinion regarding the addition of references on line 130. In the revised manuscript line 130 becomes 149. We included references [44,46] on behalf of Quang, D.N.; Ngan, V.H.; Tam, H.S.; Viet, N.T.; Tinh, N.X.; Tanaka, H. Long-Term Shoreline Evolution Using 726 DSAS Technique: A Case Study of Quang Nam Province, Vietnam. J. Mar. Sci. Eng. 2021, 9, 1124 and Nithu R.; Gurugnanam, B.; Sudhakar, V.; Francis, P.G. Estuarine Shoreline Change Analysis Along the Ennore 730 River Mouth, South East Coast of India, Using Digital Shoreline Analysis System. Geod. Geodyn. 2019, 10, 205-212.
Reviewer: L206: There is no Figure 4D. Please correct this.
Response: we have revised this section by removing Figure 4C and Figure 4D in the text. Previously, there was an error in writing so that Figure 4C and Figure 4D appeared in the text but there were no figures. We would like to confirm that only Figure 4A and Figure 4B are in this manuscript.
Reviewer: L293: Why was there strong accretion at the Playa beach and retreat at the Simeto River mouth? Do you have some explanation?
Response: Any explanations were discussed in the Results paragraph. The reason why such significant seaward migration was observed in correspondence with the Playa beach and huge landward movements were detected in correspondence with the Simeto River mouth is shown within the Discussion paragraph at lines 480-497.
Reviewer: Section 4.1. The authors are commenting on beach changes that are mostly in the range 20-30 m for the period 1972-2021 (about 0.5 m/year) but the resolution of the satellite images is less than the aforementioned changes, for example Landsat 1-5 -> 60-80 m or Landsat 8-9 -> 30-100 m. How reliable are these observations?
Response: The reliability of the shoreline extraction from the satellite images was checked by comparing shoreline positions with the shorelines obtained from high-resolution orthophotos (50-100 cm; https://www.sitr.regione.sicilia.it/; http://www.pcn.minambiente.it/mattm/, accessed on 26th April 2022) and significant overlapping between the two datasets was found. (L181-185).
Reviewer: Section 5.2. The stated RMSE are rather large when compared to the changes that are predicted. If the common error (RMSE) is of the order of magnitude (up to 116 m in the NS) of the forecasted changes (for example, 114.9 m in the Acquicella area). What does this say about the reliability of the forecasts? Please explicitly compare and comment the accuracy of other studies with the accuracy of this study: “Shoreline prediction using the Kalman Filter model method has been applied in several places such as the coastlines of Oaxaca (Mexico), Göksu Delta and Gulf of Izmit (Turkey), Cuddalore (India), southern Kuwait, the Bay of Bengal (India), and the east coast of India.” Consider normalizing the RMSE with the average beach change to get NRMSE for each beach for easier comparison.
Response: As such, the Catania coastal plain showed how the reliability of the Kalman filter model is quite higher in correspondence with straight coastal subsectors, where the RMSE assessment showed a lower value than 60 m. Indeed, the highest RMSE values were detected in correspondence with the Simeto River mouth, which was recognized as a highly dynamic coastal area, and the Playa beach subsector where the coastline showed a slight curvilinear morphology (L628-633).
"Please see the attachment"

Round 2
Reviewer 2 Report
OK by me, now